# A topographically-controlled tipping point for complete Greenland ice sheet melt

Michele Petrini[1,2], Meike D.W. Scherrenberg[3], Laura Muntjewerf[4,5], Miren Vizcaino[6], Raymond Sellevold[7], Gunter R. Leguy[8], William H. Lipscomb[8], and Heiko Goelzer[1]

[1]NORCE Norwegian Research Centre, Bjerknes Centre for Climate Research, Bergen, Norway
[2]Associate to the National Institute of Oceanography and Applied Geophysics (OGS), Trieste, Italy
[3]Institute for Marine and Atmospheric research Utrecht (IMAU), Utrecht, Netherlands
[4]Dynamic People B.V., Amsterdam, the Netherlands
[5]Royal Netherlands Meteorological Institute (KNMI), De Bilt, Netherlands
[6]Delft University of Technology (TUDelft), Delft, Netherlands
[7]Agder Energi, Kristiansand, Norway
[8]NSF National Center for Atmospheric Research (NCAR), Boulder (CO), US

**Correspondence:** Michele Petrini (mpet@norceresearch.no)

**Abstract.**

A major impact of anthropogenic climate change is the crossing of tipping points, which may have severe consequences such as the complete mass loss of the Greenland ice sheet (GrIS). At present, the GrIS is losing mass at an accelerated rate, largely due to a steep decrease in its surface mass balance (SMB, the balance between snow accumulation and surface ablation from melt and associated runoff). Previous work on the magnitude and nature of a threshold for GrIS complete melt remains controversial. Here, we explore a potential SMB threshold for complete melt of the GrIS, the impact and interplay of surface melt and glacial isostatic adjustment (GIA) in determining this threshold, and whether the GrIS exhibits characteristics commonly associated with tipping points, such as sensitivity to external forcing. To this end, we force the Community Ice Sheet Model v.2 (CISM2) by cycling different SMB climatologies previously calculated at multiple elevation classes with the Community Earth System Model v.2 (CESM2) in a two-way coupled CESM2/CISM2 transient simulation of the global climate and GrIS under high $CO_2$ forcing. The SMB calculation in CESM2 has been evaluated with contemporary observations and high-resolution modelling and includes an advanced representation of surface melt and snow/firn processes.

We find a positive SMB threshold for complete GrIS melt of 230±84 Gt/yr, corresponding to a 60% decrease in SMB and to a global mean warming of +3.4 K compared to pre-industrial CESM2/CISM2 simulated values. In our simulations, a small change in the initial SMB forcing (from 255 Gt/yr to 230 Gt/yr) and global mean warming above pre-industrial (from +3.2 K to +3.4 K) causes an abrupt change in the GrIS final volume (from 50% mass to nearly complete deglaciation). This nonlinear behaviour is caused by the SMB-elevation feedback, which responds to changes in surface topography due to surface melt and GIA. The GrIS is tipping from $\sim 50\%$ mass toward nearly complete melt when the impact of melt-induced surface lowering outweighs that of GIA-induced bedrock uplift and the (initially positive) SMB becomes and remains negative for at least a few thousand years. We also find that the GrIS is tipping toward nearly complete melt when the ice margin in the central west unpins from a coastal region with high topography and SMB. We show that if we keep the SMB fixed (*i.e., no SMB-elevation*

*feedback*) in this relatively confined region, the ice sheet retreat is halted and nearly complete GrIS melt is prevented even though the initial SMB forcing is past the threshold. Based on the minimum GrIS configuration in previous paleo-ice sheet modelling studies, we suggest that the surface topography in the central west might have played a role in preventing larger GrIS loss during the last interglacial period ~130-115 kyr BP.

## 1 Introduction

The Greenland ice sheet (GrIS) is the largest freshwater reservoir in the Northern Hemisphere, currently storing 7.42 m of sea level equivalent (SLE, Morlighem et al., 2017). At present, the GrIS is losing mass at an accelerated pace, with ice loss rates over the last decade up to six times higher than in the 1980s (Mouginot et al., 2019). This is due to both atmospheric and ocean warming, causing enhanced surface melt and runoff, and increased ice discharge to the ocean (Mouginot et al., 2019). While until the late 1990s ice discharge has been the main source of ice loss (Shepherd et al., 2012), over the last two decades the contribution of surface processes has continuously increased, and as of today, 50% of the overall GrIS mass loss increase is due to a decrease in surface mass balance (SMB, *i.e.*, the balance between snow accumulation and surface ablation from melt and associated runoff) of around 160 Gt/yr (Mouginot et al., 2019; The IMBIE Team, 2020). In the future, the GrIS is expected to continue losing mass, with its contribution to global mean sea level (GMSL) largely depending on the amount of greenhouse gas forcing to the climate system (0.01-0.10 m under SSP1-2.6, 0.09-0.18 m under SSP5-8.5 relative to 1995–2014, see Fox-Kemper et al., 2021, and references therein). Regardless of the emissions pathways, there is broad agreement that the GrIS mass loss over the 21[st] century and beyond will be dominated by SMB, while the influence of ice discharge to the ocean will diminish as the marine margins retreat to higher grounds (Robinson et al., 2012; Fürst et al., 2015; Goelzer et al., 2020; Gregory et al., 2020; Payne et al., 2021). Most of the projected decrease in SMB at the end of the century is attributed to an increase in surface melt and to the loss in refreezing capacity of the firn layer (Fettweis et al., 2013; Vizcaino et al., 2015; Sellevold and Vizcaíno, 2020; Noël et al., 2022).

As ice sheets are known to respond to surface changes over centuries and millennia rather than decades, an extremely important question is whether over longer timescales ongoing and projected 21[st] century SMB changes could lead to complete GrIS melt, possibly through self-perpetuating mechanisms triggering an abrupt ice sheet response after crossing a tipping point (*i.e.*, when a small change in forcing triggers a strongly nonlinear response in the internal dynamics of a system, qualitatively changing its future state, Lenton (2011)). When an ice sheet begins to thin and retreat at its margins due to surface melt, the positive SMB-elevation feedback is a major source for long-term mass loss: as the surface topography lowers due to ice thinning, the near surface air temperature warms and leads to further ice melt. A number of paleo and future ice sheet modelling studies agree in showing that this feedback is a self-amplifying mechanism that can drive GrIS partial or near-complete melt over multi-millennia (Robinson et al., 2012; Levermann et al., 2013; Plach et al., 2019; Gregory et al., 2020). While in this context ice loss can be further accelerated by the decrease in surface albedo associated to surface melt and ice retreat (SMB-albedo feedback: Box et al., 2012; Goelzer et al., 2016; Zeitz et al., 2021), other important negative feedbacks are also at play. Increasing surface temperatures and melt over the GrIS can be counterbalanced by larger snowfall and cloudiness

as the ice sheet margins moves towards the interior owing to regional atmospheric circulation changes, orographic effects, and the larger relative humidity of warmer air masses Gregory et al. (2020); Fyke et al. (2018). Another important process when investigating GrIS melt over multi-millennial timescales is glacial isostatic adjustment (GIA), which is the solid Earth's viscoelastic response to ice sheet unloading (Wake et al., 2016). Ice sheet thinning and retreat at the margins is followed, typically within a few centuries or millennia, by a GIA-induced bedrock uplift, with uplift rates and timing depending on the properties of the underlying lithosphere and astenosphere (Milne et al., 2018). GIA-induced uplift can therefore limit, and in some cases counterbalance, the effect of the positive SMB-elevation feedback during ice sheet retreat. A recent study showed that asynchronous changes in GrIS surface topography due to surface melt and GIA can cause complex ice sheet regimes under sustained warming, including self-sustained oscillations or partial recovery after deglaciation (Zeitz et al., 2022).

To this date, a few studies attempted to assess thresholds for complete GrIS melt. Robinson et al. (2012) showed that while a negative integrated SMB is a sufficient condition for GrIS complete melt, not accounting for the evolving ice sheet topography and SMB-elevation feedback (as done for example in Gregory and Huybrechts, 2006; Noël et al., 2021) is likely to result in an overestimation of the temperature threshold for ice sheet loss. Modelling studies including this mechanism (either explicitly or parametrised) estimated a Global Mean Temperature (GMT) threshold for GrIS complete melt over the next millennia between 1.6 K and 3 K above pre-industrial levels, with the rate of GrIS decline depending on the amount of warming above the threshold (Robinson et al., 2012; Levermann et al., 2013; Clark et al., 2016; Van Breedam et al., 2020; Gregory et al., 2020; Zeitz et al., 2022; Höning et al., 2023; Bochow et al., 2023). Paleo data and modelling indicate that during the last interglacial period (LIG, ∼130-115 kyr BP), it is unlikely that the GMT exceeded 2 K above pre-industrial levels (Capron et al., 2014), and the GrIS contributed no more than 4 m to GMSL (Helsen et al., 2013; Goelzer et al., 2016; Plach et al., 2019; Sommers et al., 2021). This suggests that during the LIG the threshold for complete GrIS melt had not been passed, or the warming had been too little or too short to induce a full GrIS collapse. Another possible paleo-analogue for 21[st] century warming is the mid-Pliocene Warm Period (mPWP, ∼3.264-3.025 Myrs BP, Haywood et al., 2011), which is estimated to have been 2-3 K warmer than pre-industrial. GrIS simulations for this period yield extremely variable ice sheet configurations, ranging from an ice-free state to a near modern GrIS depending on the climate forcing selected (Dolan et al., 2015). Altogether, paleo and future modelling studies both suggest the existence of a threshold for GrIS complete melt in a warmer climate. However, there is still uncertainty on the nature of this threshold, and whether or not the GrIS would exhibit a tipping point behaviour once this threshold is passed. A recent study using an ice sheet model bi-directionally coupled to a low-resolution atmosphere Global Circulation Model (GCM) found a gradual, almost linear decline of GrIS equilibrium mass in response to sustained global mean warming (Gregory et al., 2020), rather than the sharp threshold behaviour found in earlier work (Robinson et al., 2012; Levermann et al., 2013). A better understanding of the mechanisms regulating the GrIS response to sustained warming is needed to determine the existence of tipping points for GrIS complete ice sheet melt, and as such reduce uncertainties on the GrIS long-term sea level rise (SLR).

Here, we explore a potential SMB threshold for complete melt of the GrIS, the impact and interplay of surface melt and glacial isostatic adjustment (GIA) in determining this threshold, and whether the GrIS exhibits characteristics commonly associated with tipping points, such as sensitivity to external forcing. To this end, we force the higher-order Community Ice Sheet

Model v.2 (CISM2, Lipscomb et al., 2019) with different SMB climatologies previously calculated with the full-complexity Community Earth System Model v.2 (CESM2, Danabasoglu et al., 2020) in a two-way coupled CESM2/CISM2 simulation of the global climate and GrIS (Muntjewerf et al., 2020b). The SMB calculation in CESM2 is based on a surface energy balance calculation, and includes an advanced representation of snow/firn processes, such as snow compaction, refreezing, and surface albedo (Muntjewerf et al., 2021; Sellevold and Vizcaíno, 2020). In our simulations, the SMB forcing is updated as the GrIS surface elevation evolves through an elevation classes calculation in the land component of CESM2 (Muntjewerf et al., 2021), while bedrock changes due to glacial isostatic adjustment (GIA) effect are accounted for in CISM2 through the Elastic Lithosphere-Relaxing Asthenosphere (ELRA) method (Le Meur and Huybrechts, 1996; Rutt et al., 2009).

The paper is structured as follows: in Section 2, we briefly describe CISM2 (2.1), the SMB forcing (2.2) and the experimental setup (2.3). In Section 3 we analyse the main results, which are organised in three subsections on (3.1) thresholds, (3.2) pattern and timescales, and (3.3) processes for GrIS complete melt. In Section 4 we first discuss the thresholds found here in the context of the existing literature and end-of-century projections (4.1), and the existence of a tipping point for GrIS decay (4.2). We then draw our main conclusions and suggest possible future research directions (4.3).

## 2 Method

### 2.1 Ice sheet model description

CISM2 is a parallel, three-dimensional thermomechanical ice sheet model (ISM) which solves equations for the conservation of mass, momentum, and internal energy (Lipscomb et al., 2019). In this study, CISM2 solves a depth-integrated viscosity approximation of the Stokes equations for incompressible viscous flow (DIVA, Goldberg, 2011). At the horizontal resolution used in this study for the Greenland domain (4 km), the DIVA outperformed other hybrid or zero-order solvers in terms of both model performance and representation of the ice-flow physics (Robinson et al., 2022). Basal sliding is calculated using a pseudo-plastic sliding law which incorporates linear, plastic and power-law behaviour (Aschwanden et al., 2016). Ocean forcing at the marine-terminating outlet glaciers is not accounted here, and floating ice is immediately removed based on a simple flotation criterion. GIA is parametrised using the Elastic-Lithosphere/Relaxing-Asthenosphere (ELRA) model (see for example Rutt et al., 2009), which allows to take account for a regional bedrock response to changes in ice load. The main CISM2 parameter values used in this study are provided in Table A1. Finally, while a more detailed description of all the CISM2 features and options can be found in Lipscomb et al. (2019), we highlight that the same model setup used here has been previously applied in two-way coupled simulations of the global climate and the GrIS over the next centuries (Muntjewerf et al., 2020b, a) as well as in the past (Sommers et al., 2021).

### 2.2 SMB forcing

Here, we provide a brief description of the general SMB calculation in CESM2 and how it is downscaled onto the CISM2 grid.

The SMB calculation is done in the land component of CESM2 (Community Land Model v.5, CLM5 Lawrence et al., 2019) and is defined as:

$$SMB = Precipitation - Runoff - Sublimation;$$
$$Precipitation = Rain + Snowfall;$$
$$Runoff = Rain + Melt - Refreezing.$$

Snowfall is calculated from the total precipitation depending on the surface temperature (100% snowfall if the temperature is below -2 °C, 100% rain if the temperature is above 0 °C, and linear interpolation in between). Rainfall can contribute positively to the SMB if it refreezes within the snowpack. Snow and ice melt are calculated based on the melt energy available within the ice column, which depends on the sum of net surface radiation, latent and sensible turbulent surface fluxes, and ground heat fluxes at the atmosphere-snow interface (Lawrence et al., 2019). In order to overcome the challenge of resolving steep SMB gradients around the ice sheet margins at relatively coarse typical CLM5 resolutions (1° in Muntjewerf et al., 2020b), the SMB is calculated at 10 elevation classes, with boundaries at 0, 200, 400, 700, 1000, 1300, 1600, 2000, 2500, 3000, and 10000 m. This is done first by downscaling the CLM5 grid cell temperature to each elevation class using a uniform lapse rate of -6 K/km. A vertically uniform relative humidity is then assumed to determine the potential temperature, specific humidity, air density, and surface pressure over each elevation class. The interpolated fields are then used to calculate the SMB components in each elevation class. After annual mean SMB is calculated at each elevation class in CLM5, it is then downscaled to the higher-resolution CISM2 domain through horizontal bilinear interpolation and linear vertical interpolation between adjacent elevation classes to the ice sheet model surface elevation. Remapping onto the CISM2 grid is followed by a normalization step that guarantees conservation of SMB fluxes in both the accumulation and ablation zones. More details on the SMB calculation and on the CESM2/CISM2 coupling are also provided in Muntjewerf et al. (2021).

## 2.3 Experimental setup

In this study, we use a functionality of CESM2 that allows to run CISM2 within the Earth System Model (ESM) architecture, forced with coupler history files of annual mean SMB calculated in previous CESM2 simulations at multiple elevation classes. Under this setup, each coupler history file contains annual mean SMB values at multiple elevation classes on the 1°land model (CLM5) grid in CESM2. During runtime, every year the CESM2 coupler selects an individual coupler history file and performs the trilinear (bilinear horizontal + linear vertical) interpolation of the annual mean SMB to the GrIS surface elevation on the CISM2 4 km grid, which is then followed by the normalization step. In this way, the SMB is automatically updated during runtime for changes in ice sheet geometry, thus allowing us to account for the SMB-elevation feedback based on an advanced energy balance calculation accounting for snow/firn processes and energy fluxes at the ice sheet surface. We highlight that this setup allows the use of SMB forcing files calculated in previous CESM2 simulations, and as such changes in ice sheet geometry and melt are not feeding back on the climate (*i.e.*, we keep CESM2 in data mode while using its coupler functionality). Our CISM2 simulations are forced with multiple-elevation class SMB climatologies from a published, two-way coupled CESM2/CISM2 simulation of the global climate and GrIS under an idealized, high Greenhouse Gas (GHGs)

scenario (hereinafter referred to as BG-1pct run, Muntjewerf et al., 2020b). In this simulation, atmospheric $CO_2$ concentration is increased by 1% every year until it reaches four times pre-industrial values at year 140, after which it is kept fixed. From the BG-1pct run, we select several 23-years climatologies corresponding to different SMB and global mean temperature levels (values are shown in Table 1, see also Fig.1a). The choice of each climatology being 23 years-long results as a compromise between keeping the SMB standard deviation within the range of pre-industrial values and a sampling rate large enough to prevent data aliasing. In each simulation, we cycle the 23-years, multiple-elevation classes SMB forcing for 80 kyr (or less if the ice sheet disappears earlier).

Each run is restarted from the BG-1pct run at the last year of the corresponding forcing interval (initial and final year of each forcing interval are shown in Table A2). This means that model parameter choices and the simulated GrIS thermodynamic memory rely on the spin-up of the coupled BG-1pct simulation, which is a two-way coupled CESM2/CISM2 pre-industrial equilibrium simulation of the global climate and GrIS described in (Lofverstrom et al., 2020). In this spin-up, Lofverstrom et al. (2020) combine fully and partially coupled model configurations, running the (Greenland) ice sheet model component for 10 kyr. In the spin-up, the GrIS internal temperature was initialized with a 3-D temperature structure corresponding to the 9 kyr BP GrIS state in a full glacial cycle ice sheet model simulation (Fyke et al., 2014). This means that at the end of the spin-up simulation the GrIS internal temperature and rheology account for the thermal memory of the last glacial cycle. The ice sheet is free to evolve beyond its observed present-day extent throughout the 10 kyr of CISM2 simulation, and at the end of the spin-up procedure GrIS volume and area are overestimated by 12% and 15%, respectively, compared to present-day observations (Lofverstrom et al., 2020). During the spin-up procedure, model parameters (see Table A1) have been selected to best match present-day observations of ice extent, thickness, and velocity (Lofverstrom et al., 2020).

In addition to the main simulations presented in this study, we performed several sensitivity simulations in which:

– we switch off the GIA parametrisation in CISM2, to isolate the impact of GIA on the GrIS response to sustained warming;

– we test different values for two model parameters in the GIA parametrisation: (i) characteristic mantle relaxation time, (ii) lithosphere flexural rigidity. The former determines the timing of the viscoelastic response of the astenosphere (Cathles et al., 2023), whereas the latter embodies the lateral stiffness of the lithosphere (Walcott, 1970). For each parameter, we test lower and higher-end values following previous work by Zweck and Huybrechts (2005) (see Table A1). With these sensitivity tests we aim to assess the dependence of the SMB thresholds and tipping behaviour of the GrIS on the selected GIA model parameters;

– we prescribe lower values of the minimum friction angle in the pseudo-plastic sliding law (see Table A1). These sensitivty tests are aimed at assessing the impact of increasing sliding velocities (and, as consequence, increased ice discharge) on the thresholds and tipping behaviour of the GrIS.

## 3   Results

Here we first describe the relationship between SMB forcing and final GrIS volume, to provide an SMB threshold for GrIS
complete melt. We then illustrate the pattern and timescales for ice retreat, in particular for simulations close to the SMB
threshold. We conclude by assessing the processes that determine the GrIS response to sustained warming.

### 3.1   Thresholds for GrIS complete melt

In our simulations, the GrIS exhibits a sharp threshold behaviour, with a nonlinear relationship between initial SMB forcing
and GrIS melt/mass loss (Fig. 1b). In Table 1 we provide the values of initial SMB forcing, corresponding transient global mean
and GrIS warming above pre-industrial, and resulting final GrIS volume for each simulation. Final GrIS states are clustered in
three main groups: (a) "low melt", *i.e.*, less than 25% GrIS mass loss; (b) "medium melt", *i.e.*, around 50% GrIS mass loss,
and (c) "complete melt", *i.e.*, more than 80% GrIS mass loss. GrIS limited loss (<25%) is obtained for an initial SMB forcing
above 317±97 Gt/yr. Compared to the two-way coupled CESM2/CISM2 pre-industrial equilibrium simulation (hereafter BG-
PIcontrol run, Lofverstrom et al., 2020, 585±85 Gt/yr, see Table 2), this first SMB threshold for "low melt" corresponds to a
decrease of the GrIS integrated SMB from pre-industrial not exceeding 40% (the GrIS pre-industrial SMB in the BG-PIcontrol
run is 585±85 Gt/yr), and to a transient global mean warming above pre-industrial below 2.8 K. An ice loss of around 50%
of the initial GrIS mass is obtained for an intermediate initial SMB forcing between 286±94 and 255±83 Gt/yr (between 40
and 50% decrease from the pre-industrial BG-PIcontrol SMB), corresponding to global mean warming above pre-industrial
between 2.8 and 3.4 K. Finally, we find that the GrIS is bound to completely melt if the initial SMB forcing is lower than
230±84 Gt/yr (Fig. 1b and Table 1). This corresponds to a decrease of around 60% of the GrIS integrated pre-industrial SMB
at the end of the BG-PIcontrol run. The SMB threshold for complete GrIS melt corresponds to, compared to the BG-PIcontrol
run, a transient global mean warming above pre-industrial of 3.4 K (Fig. 1a and Table 1).

### 3.2   Pattern and timescales of GrIS retreat

In the "low melt" simulations, ice loss mostly takes place at the southwestern margin, with limited retreat in the north occurring
in the +2.8 K run (see inset in Fig. 1b). The southwestern GrIS margin retreats all the way to the east in the "medium melt"
simulations, until a large ice cap in the southern tip of Greenland is disconnected from the GrIS main body. In these simulations,
showing an intermediate GrIS loss, significant retreat also occurs in the north, whereas the midwestern ice sheet margin remains
close to the coast (see inset in Fig. 1b). When the SMB threshold for complete GrIS melt is passed, the GrIS retreats all the
way towards the east, with isolated ice caps of variable size remaining in regions of high bedrock elevation (see inset in Fig.
1b). Timescale needed to reach ice sheet deglaciation or stabilization increases sharply as the initial SMB forcing gets close
to the threshold for GrIS complete mass loss. In the "low melt" simulations, minimum GrIS volume is reached within the first
15 kyr (Fig. 2 and Table 1). Timescales are doubled for the "medium melt" simulations, in which the GrIS attains minimum
volume after around 30-35 kyr. The simulation immediately above the SMB threshold (+3.4 K run) shows the longest response
time, as the GrIS keeps decreasing until 40 kyr, after a retreat slowdown between 15 and 20 kyr (Fig. 2 and Table 1). For the

other simulations in the "complete melt" group, timescales decrease quickly as the initial SMB forcing gets lower and moves away from the threshold. For a positive initial SMB forcing, the shortest GrIS deglaciation time is around 12 kyr, whereas if the initial SMB forcing is negative Greenland becomes mostly ice-free in 8 kyr or less (Fig. 2). Detailed GrIS retreat pattern and timescales for each simulation are displayed in the supplementary videos.

### 3.3    Processes controlling ice retreat

In our simulations, the response of the GrIS to sustained warming is primarily determined by the interplay of surface melt and GIA, which are interconnected through the SMB-elevation feedback. For every initial SMB forcing level, the ice sheet loses mass during the first 10-15 kyr, and the SMB further decreases due to ice thinning and surface elevation lowering (Fig. 2). However, the impact of ice thinning is mitigated, and in some cases counterbalanced, by GIA: while the ice sheet thins and retreats, the bedrock uplifts, thus increasing the surface elevation and, consequently, the SMB (see bottom panel in Fig.
2 and supplementary videos, for instance the video showing the +3.2 K and 3.4 K runs). This interplay is key in determining whether the ice sheet will disappear or not: GrIS complete loss is achieved only in simulations where the surface lowering outweights the bedrock uplift, and the SMB becomes and remains negative for at least a few thousand years (Fig. 2). When this does not happen, the GrIS eventually reaches a stable configuration as in the "low melt" or "medium melt" groups. In the simulations that are closest to the threshold for complete GrIS melt, the interplay of SMB-elevation feedback and GIA is
also causing self-sustained, quasi-periodic ice sheet oscillations ranging from 3% to 12% of the initial GrIS mass. In fact, in some cases the increase in SMB due to GIA-induced bedrock uplift can promote ice thickening and margin readvance. This mechanism is then reverted when the isostatic subsidence resulting from ice thickening causes the SMB to decrease again, enough to promote ice thinning and margin retreat (see supplementary videos of the +3.0, +3.2, and +3.4 runs). The amplitude and periodicity of these oscillations increase as the initial SMB forcing gets closer to the threshold for complete GrIS melt.
The highest oscillation amplitude and periodicity is obtained in the simulation immediately after the SMB threshold is crossed (+3.2 K run, with a volume oscillations of 12% of the initial ice mass and over cycles of 30 kyr, see Fig. A1).

    If GIA is switched off, the response of the GrIS changes drastically, as shown in Fig. 3 and 4. When we turn off GIA for a simulation in the "low melt" group (+2.5 K run), SMB-elevation feedback due to surface lowering causes the GrIS to deglaciate in 20 kyr (Fig. 3). This is equivalent, both in terms of final GrIS mass and timing response, to a simulation in the
"complete melt" group, with an initial SMB forcing above the threshold for GrIS complete melt (+3.6 K run, see Fig.2). For other simulations in the "medium melt" or "complete melt" group, removing the GIA effect results in rapid GrIS deglaciation in around 10 kyr or less (Fig.3). We find that when GIA is not accounted for, complete GrIS melt is achieved for an initial SMB forcing of 488±91 (two times higher than when GIA is included and corresponding to a global mean warming above pre-industrial of 1.6 K) and the GrIS threshold behaviour is exacerbated, with the final GrIS state switching directly from "low
melt" to "complete melt" (Fig. 4). The primary role played by GIA in determining the SMB threshold and response of the GrIS to sustained warming also emerges in sensitivity tests repeating the first simulation after the SMB threshold is passed (+3.4 K run) under lower and higher-end values for model parameters in the GIA parametrisation (characteristic mantle relaxation time and lithosphere flexural rigidity, see Subsection 2.3 and Table A1). While in these sensitivity experiments the GrIS loses more

than 80% of its initial mass for all the parameter values explored, timescales to reach minimum ice sheet volume, as well as the timing and strength of GIA-induced oscillation vary considerably (Fig. 5). In particular, for a low mantle relaxation time, GrIS deglaciation is slower and more gradual. For a high value of the characteristic mantle relaxation time, the GrIS deglaciates faster (around 15 kyr) and the ice volume remains low until 50 kyr. However, after that time the GrIS regrows up to 40% of its initial volume. A similar late regrowth is also observed for a low value of the lithosphere flexural rigidity, although in this case minimum GrIS volume is reached much later between 60 and 70 kyr.

Finally, we tested the impact of increased ice discharge on the long-term GrIS evolution. As in our simulations we do not account for ocean forcing at the marine-terminating outlet glacier, in our sensitivity simulations we triggered an increase in ice discharge by prescribing lower values of the minimum friction angle in the pseudo-plastic sliding law, see Subsection 2.3). In particular, we repeated the simulation immediately below the SMB threshold for complete GrIS melt (+3.2 K) imposing an extremely low friction angle value (0.5°, see green dashed curve in Fig. A3). In this sensitivity test, we immediately observe an increase in ice discharge due to higher sliding velocities, and after 500 years in the simulation the total GrIS mass budget is the same as in the 3.8 K run (which is well above the SMB threshold, see Fig. 2). Nevertheless, the long-term evolution of the +3.2 K simulation under increased sliding is almost identical to the reference case, with a final GrIS volume of around 50% of its pre-industrial value. This shows that in our simulations the GrIS long-term response to sustained warming and the SMB threshold for complete GrIS melt are dominated by surface processes.

### 3.4 Topographic control on GrIS complete melt

A comparison of the simulations immediately below and above the SMB threshold for GrIS complete melt (+3.2 K and +3.4 K runs, respectively) indicates that the ice margin position in the central west is crucial to determine the final ice sheet state. When the GrIS central western margin remains sufficiently close to a coastal region with relatively high topography and SMB (indicated by the red box in Fig. 6), possibly reestablishing connection during margin readvances within self-sustained oscillations, the ice loss in the southwest and north remains limited (see +3.2 K run in Fig. 6 and relative supplementary video). When instead the midwestern margin permanently loses contact with this coastal region, the GrIS retreats towards the east and loses more than 80% of its mass (see +3.4 K run in Fig. 6 and relative supplementary video). In the first simulation above the SMB threshold, the central western margin starts to retreat towards the east only after 20 kyr, when the connection between the ice sheet and the high topography coastal region is permanently lost. In the remaining simulations after the SMB threshold is crossed (+3.6 K, +3.8 K, +4.0 K, +4.5 K, +5.2 K), the connection between the central western margin and the coastal region with high topography is lost earlier (around 15 kyr or sooner) and the retreat pattern is more uniform in time (see right panel in Fig. 6 and relative supplementary videos). To further test the role played by the surface topography in the central west on the GrIS abrupt retreat, we repeated the first simulation above the SMB threshold (+3.4 K run) while keeping the SMB fixed (*i.e.*, inhibiting the SMB-elevation feedback) in the high topography coastal region (*i.e.*, inside the red box in Fig. 7). When we impose this (local) additional constraint, GrIS nearly complete mass loss is prevented, with a final ice sheet extent (right panel in Fig. 7) and volume (Fig. 8) almost identical to those of the simulation immediately below the SMB threshold (+3.2 K run). GrIS complete loss is also prevented when we apply the same local SMB-elevation feedback constraint to the +3.6 K

simulation (in which the GrIS is deglaciating in 20 kyr), whereas for higher forcings GrIS loss cannot be prevented. Overall, our simulations show that the relatively confined, central western coastal region has a stabilizing effect on the GrIS response to sustained warming, and as such plays a primary role in determining whether the ice sheet will or will not pass or not the threshold towards complete melt.

## 4 Discussion

In this section, we first compare the thresholds found here with the existing literature and we will analyze these thresholds in the context of end-of-century projections. Then, we examine the behaviour of the GrIS with a specific focus on the potential tipping point associated with the SMB threshold for complete melt. While we discuss the GrIS response to sustained melt, the non-linear and abrupt nature of those responses and the sensitivity to external factors, it is important to reiterate here that our primary objective is to explore the SMB threshold itself, rather than the aspects of multistability or reversibility often associated with tipping points.

### 4.1 Thresholds for complete GrIS melt

Here we have found a positive SMB threshold of 230±84 Gt/yr for complete GrIS melt. This corresponds to a 60% decrease from the GrIS pre-industrial SMB calculated in a two-way coupled CESM2/CISM2 equilibrium simulation (585±83 Gt/yr, see Table 2 and Lofverstrom et al., 2020). In this pre-industrial two-way coupled ESM/ISM simulation no corrections are applied to the SMB (*e.g.*, SMB masking outside the present-day GrIS extent) and the GrIS is left free to evolve. Although CESM2 compares reasonably well with ERA-Interim and RACMO2 for several key controls on the SMB over Greenland (surface melt, runoff, longwave radiation van Kampenhout et al., 2020; Noël et al., 2020), this approach led to an overestimation of the GrIS initial extent (15%), volume (13%) and, as a consequence, to an integrated SMB over the GrIS which is higher than in uncoupled CESM2 experiments of the historical period (390±28 Gt/yr, see Table 2 and Lofverstrom et al., 2020) and recent satellite observations (437±17 Gt/yr, Mouginot et al., 2019). In view of this, the positive SMB threshold found here is coupled-model dependent, and it remains complicated to analyse in the context of end-of-century projections, or to make a direct comparison with other modelling studies and recent observations. Constraining ISM simulations to GrIS changes during the LIG, Robinson et al. (2012) found a positive SMB threshold for complete ice sheet melt ranging between 150-340 Gt/yr. This agrees reasonably well with our study, although in Robinson et al. (2012) the pre-industrial SMB is lower (around 400 Gt/yr). In the two-way coupled CESM2/CISM2 projection for the SSP5-8.5 scenario, the SMB crosses our threshold for GrIS instability between years 2060-2080, and by the end of the century it has become negative (Muntjewerf et al., 2020a). In our simulations, a negative SMB forcing leads to a relatively rapid GrIS deglaciation in less than 10 kyr (Table 1 and Fig. 2). While coupled CESM2/CISM2 simulations for lower emission scenarios (SSP1-2.6, SSP2-4.5, SSP3-7.0) are currently not available, in the corresponding CESM2-only future projections the SMB over the GrIS at the end of the century is well below the threshold found here, regardless of the emission scenario considered (see Table 2). This comparison, however, must be

considered carefully as (a) the GrIS pre-industrial SMB is lower in CESM2-only runs, (b) in CESM2-only runs, the SMB is calculated over a fixed present-day GrIS topography, and as such the SMB-elevation feedback is not accounted for.

The SMB threshold found here corresponds to a transient global mean and GrIS warming of 3.4 K and 3.2 K above pre-industrial, respectively. Our global mean warming threshold is at the high end of previous ice sheet modelling studies with climate forcing (a) calculated offline with GCMs (3.1 K, Gregory and Huybrechts, 2006) (b) interactively calculated with regional models or intermediate complexity ESM (1.6-3 K, Robinson et al., 2012; Levermann et al., 2013; Gregory and Huybrechts, 2006; Höning et al., 2023). We highlight however that in our forcing method the SMB has not fully equilibrated with the climate, and a more sustained warming of 3.4 K above pre-industrial would likely result in a lower SMB as the snow pack deteriorates (ablation area expansion, less refreezing capacity). To properly determine a global mean warming threshold for complete GrIS melt would require running further CESM2/CISM2 fully or asynchronously coupled experiments, which is computationally too expensive and out of the scope of this study.

## 4.2 GrIS tipping point behaviour

Our simulations indicate that the GrIS response to sustained warming is nonlinear, and that the ice sheet is tipping from 50% mass towards complete melt once the positive SMB threshold is passed. This agrees well with early modelling studies (Gregory and Huybrechts, 2006; Robinson et al., 2012; Levermann et al., 2013). However, our results do not agree with Gregory et al. (2020), which suggests a gradual, almost linear GrIS decline in response to sustained warming. In their study, Gregory et al. (2020) used an ISM bi-directionally coupled to a low-resolution atmosphere GCM, and used elevation classes to downscale the SMB onto the ice sheet model grid. While in our simulations we also use elevation classes to downscale the SMB forcing onto the CISM2 grid, GrIS changes are not communicated back to CESM2, and as such we do not account for a number of ice-atmosphere feedbacks (*e.g.*, increased cloudiness and snowfall as the ice sheet margin moves towards the interior). In Gregory et al. (2020), these processes result in an increase in precipitation which in some cases counterbalances the effect of the SMB-elevation feedback and leads to the multiple stable GrIS states. However, in our simulations the GrIS tipping behaviour is determined by the interplay of SMB-elevation feedback and GIA, with the latter not being accounted for in Gregory et al. (2020). While we cannot exclude that accounting for ice-atmosphere feedbacks would stabilize the GrIS and change the SMB threshold, it is also difficult to determine if they would drastically change the nonlinear nature of the GrIS response to sustained warming found here. Another recent study by Zeitz et al. (2022) found a similar, nonlinear impact of the interplay between GIA and SMB-elevation on the GrIS response to sustained warming, with final GrIS states clustered in three main groups for a global warming exceeding 2 K above pre-industrial ("partial recovery", "oscillations", "loss"). Zeitz et al. (2022) used a different ISM and GIA parametrisation than in our study, thus suggesting that the GrIS tipping behaviour simulated here is unlikely to depend on our model setup. Our study also agrees well with Zeitz et al. (2022) and Plach et al. (2019) in showing that increased ice discharge rates have no effect on the long-term GrIS equilibrium response to sustained warming.

We find that the GrIS is tipping towards complete melt if the ice margin in the central west retreats enough to disconnect from a group of isolated ice caps with high bedrock elevation and SMB. If the ice margin remains pinned to this topographic pinning point, GrIS loss is limited also in southwestern and northern Greenland, and does not exceed 50% of the initial GrIS mass.

The presence of this topographic feature of the GrIS, in combination with a stable midwestern margin position and limited ice loss in the southwest and north has been consistently simulated in ice sheet modelling studies of the LIG (Helsen et al., 2013;

Plach et al., 2019; Goelzer et al., 2016; Sommers et al., 2021). In these simulations, the midwestern margin remains pinned to the topographic pinning point, and the GrIS does not lose more than 4 m SLE. In particular, in a recently published two-way coupled CESM2/CISM2 simulation of the global climate and GrIS during the LIG, the ice sheet extent at 123 kyr ago is very similar to those in our simulations belonging to the 'medium melt' group (see Fig. 3 in Sommers et al., 2021). After this time, however, the SMB becomes positive and the GrIS starts to readvance towards the west. This suggests that around 123 kyr the

GrIS might have been close to tipping towards a much reduced state. Compared with other modelling studies indicating a GrIS tipping point behaviour in response to sustained warming, our study agrees in showing that the ice sheet retreats towards the east, and that the midwestern margin remains close to the coast before tipping (Robinson et al., 2012; Höning et al., 2023; Zeitz et al., 2022).

## 4.3 Conclusions and future research

Here, we have used a state-of-the-art, higher-order ice sheet model (CISM2) to (1) determine a SMB threshold for GrIS complete melt, (2) investigate the nature of this threshold, (3) understand the processes controlling the ice sheet response. We forced CISM2 with different levels of SMB, previously calculated at multiple elevation classes with a full-complexity Earth-System Model (CEMS2) which includes an advanced representation of snow/firn processes, surface albedo and melt. In our simulations, the SMB is updated as the ice sheet surface elevation changes through the elevation classes method. This

allows us to account for the SMB-elevation feedback during GrIS lowering and retreat. Bedrock changes due to GIA effects are accounted for in CISM2 through the ELRA parametrisation.

We have found a positive SMB threshold for complete GrIS melt of 230±84 Gt/yr, corresponding to a 60% decrease from the GrIS pre-industrial equilibrium SMB. This SMB threshold corresponds to a transient global mean and GrIS warming of 3.4 K and 3.2 K above pre-industial, respectively. The thresholds found here are in overall agreement with previous studies. In our

simulations, the response of GrIS to sustained warming is nonlinear, and final ice sheet state are clustered in three main groups: (a) "low melt" (<25% mass loss), (b) "medium melt" (∼50% mass loss), (c) "complete melt" (>80% mass loss). This tipping behaviour is determined by the effect of SMB-elevation feedback in response to surface melt and GIA. While topographic lowering due to surface melt promotes further melt and ice loss, GIA causes a bedrock uplift which in turn yields reduced melt, ice thickening and margin stabilization or readvance. The GrIS is tipping from ∼50% mass towards complete loss when

the melt-induced surface lowering outweighs the GIA-induced bedrock uplift, and the initially positive SMB becomes and remains negative for at least a few thousand years. A similar tipping behaviour of the GrIS, resulting from the interplay of SMB-elevation feedback and GIA, has also been recently simulated by Zeitz et al. (2022) using a different ISM and GIA parametrisation. However, recent two-way coupled modelling work suggests that ice-atmosphere negative feedbacks which are not accounted for in our simulations might dampen the nonlinear response of the GrIS to sustained warming (Gregory et al.,

2020). We have found that whether the GrIS will tip towards complete melt or not depends on the ice sheet margin position in the central west. When the ice margin remains close/connected to a coastal region with high bedrock elevation and SMB, GrIS

loss is also limited in the southwest and north and does not exceed ∼50% of its initial volume. Previous modelling studies of the GrIS during the LIG show that the ice sheet had, at its minimum extent, a similar configuration as in our simulations in the "medium melt" group. In view of this, we suggest that the central-western high topography coastal region might have played a similar stabilizing role on the GrIS evolution during the LIG.

The modelling approach presented here includes an advanced representation of the SMB-elevation and albedo feedback through the CESM2 melt energy calculation, snow/firn model and elevation classes approach. Nevertheless, the main caveat of this study is that CISM2 is not bi-directionally coupled to CESM2, and as such we do not account for a number of ice sheet/atmosphere feedbacks (*e.g.*, increase in cloud cover and precipitation as the GrIS retreats towards the interior). Moreover, there are uncertainties arising from the elevation classes approach which is at the core of our SMB forcing method. In CESM2, the air temperature is downscaled to each elevation class using a constant lapse rate of 6 K/km. While Sellevold et al. (2019) demonstrated that this value yields realistic SMB gradients over the GrIS, air temperature lapse rate values can show relatively large variations depending on conditions at the ice surface, seasonality or even on the background climate (Erokhina et al., 2017). The specific simulation setup used in this study, however, does not allow to explore the impact of different lapse-rate values, as the SMB at multiple elevation classes was previously calculated in a fully coupled CESM2/CISM2 simulation (Muntjewerf et al., 2020b) and is used in this study as a forcing file only. Finally, our sensitivity experiments show that the GrIS response to sustained warming might vary under different, physically realistic values of GIA parameters, especially in terms of timescales. Although widely used in ice sheet model simulations, the GIA parametrisation used in our study is relatively simple, and do not account for lateral variations of the lithospheric thickness and upper-mantle viscosity. To better understand the nature of the response of the GrIS to sustained warming, further ice sheet-climate-solid Earth coupled modelling efforts should be dedicated to fully or better resolve the interplay of ice-atmosphere feedbacks and GIA. In addition, future modelling studies of the GrIS during the LIG could shed light on how close the GrIS might have been to tipping at its minimum extent, and which processes might have prevented that.

*Code and data availability.* CESM2 is an open source model, available at https://www.cesm.ucar.edu. Computing and data storage resources, including the Cheyenne supercomputer (doi:10.5065/D6RX99HX), were provided by the Computational and Information Systems Laboratory (CISL) at NSF NCAR. Forcing files from the two-way coupled CESM2/CISM2 simulations and setup scripts to reproduce our simulations are stored in the Cheyenne supercomputer and available upon request.

*Video supplement.* Supplementary Videos can be found on Zenodo (https://doi.org/10.5281/zenodo.8384527).

*Author contributions.* MV planned the research; MS performed an initial set of simulations; MP performed the remaining simulations, analysed the output and wrote the manuscript. All authors discussed the results and reviewed and edited the manuscript.

*Competing interests.* The contact author has declared that none of the authors has any competing interests.

*Acknowledgements.* The CESM project is supported primarily by the National Science Foundation (NSF). This material is based upon work supported by the National Center for Atmospheric Research, which is a major facility sponsored by the NSF under Cooperative Agreement 1852977. MP, MV, LM, and RM have received funding from the European Research Council (Grant ERC-StG-678145-CoupledIceClim). MP and HG have received funding from the Research Council of Norway under project 324639. Storage resources were provided by Sigma2 - the National Infrastructure for High Performance Computing and Data Storage in Norway through project NS8006K. In this manuscript, we used colour schemes from Crameri (2018) to prevent visual distortion of the data and exclusion of readers with colour-vision deficiencies (Crameri et al., 2020).

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

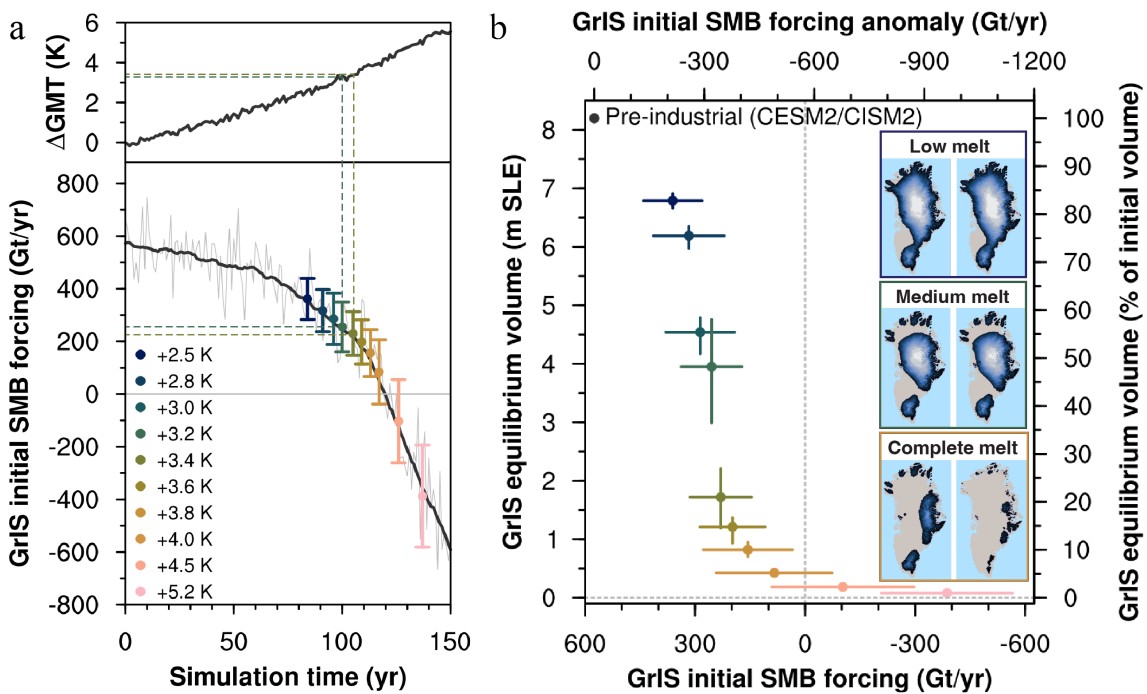

**Figure 1. (a)** Timeseries of (upper panel) Global Mean Temperature (GMT) anomaly relative to pre-industrial and (bottom panel) GrIS integrated surface mass balance in the two-way coupled BG-1pct simulation (Muntjewerf et al., 2020b). In the bottom panel, the SMB forcing used in our simulations (avg±stddev) is indicated in different colours for each run (see the inset legend). Dashed lines highlight the transition between "medium" to "complete" GrIS melt. **(b)** Scatter plot of GrIS integrated initial SMB forcing (avg±stddev, x-axis) *vs* GrIS final volume (avg±min/max, y-axis). Avg/min/max final volume are calculated within quasi-periodic oscillations, see 3.3. Colours for each simulation as in the inset legend in (a). In the inset maps, GrIS final extent is showed for two simulations belonging in each group (for the "complete melt" group, +3.2 K and +4.0 K runs are shown).

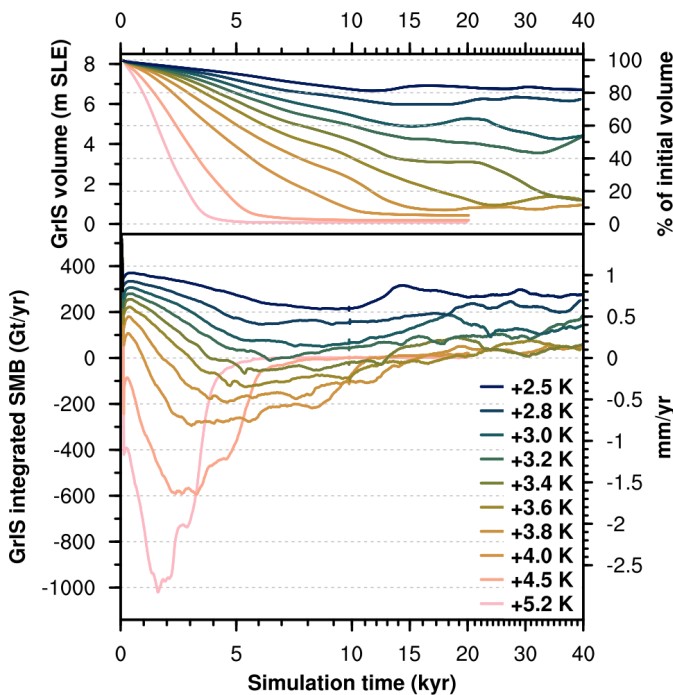

**Figure 2.** Timeseries of (upper panel) GrIS volume and (bottom panel) integrated surface mass balance for the main runs presented in this study. Note that simulation time in the x-axis is irregular (minor tickmarks every 1000 years).

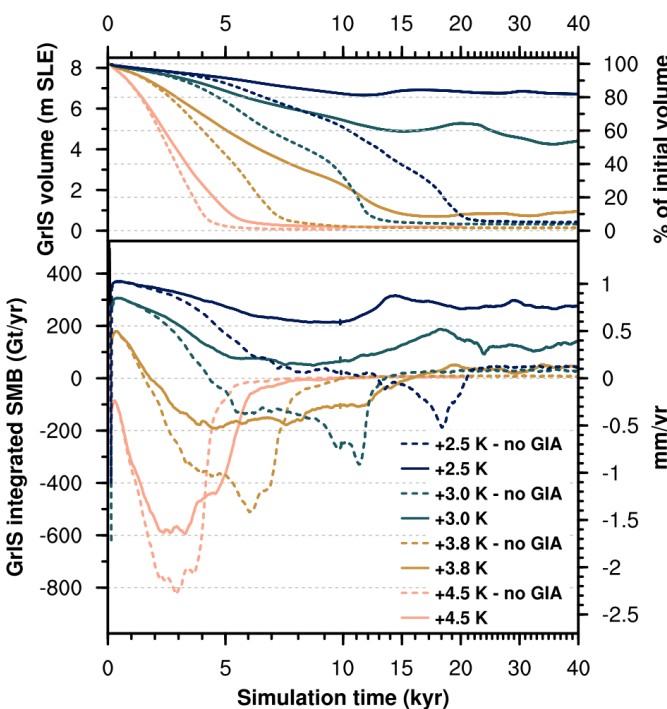

**Figure 3.** Timeseries of (upper panel) GrIS volume and (bottom panel) integrated surface mass balance for selected sensitivity simulations with GIA turned on (solid lines) and GIA turned off (dashed lines). Note that simulation time in the x-axis is irregular (minor tickmarks every 1000 years).

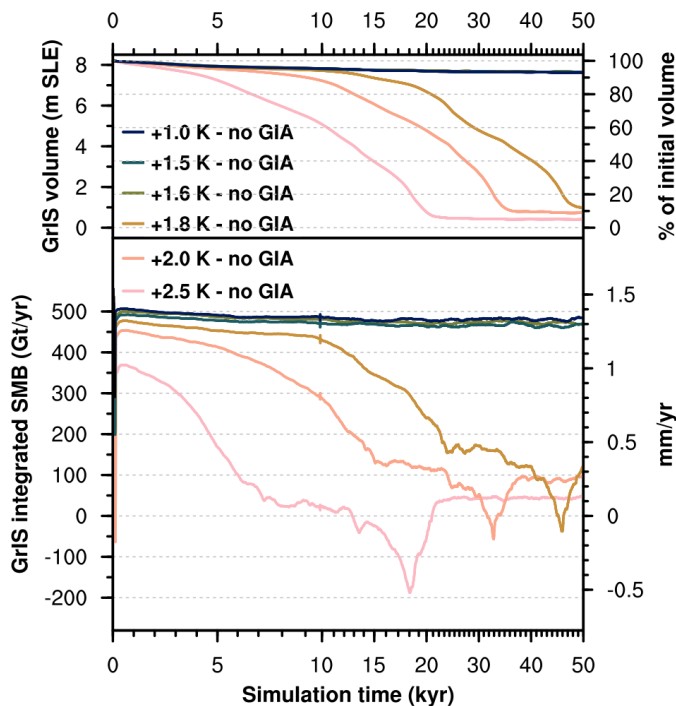

**Figure 4.** Timeseries of (upper panel) GrIS volume and (bottom panel) integrated surface mass balance for selected sensitivity simulations with GIA turned off. Note that simulation time in the x-axis is irregular (minor tickmarks every 1000 years).

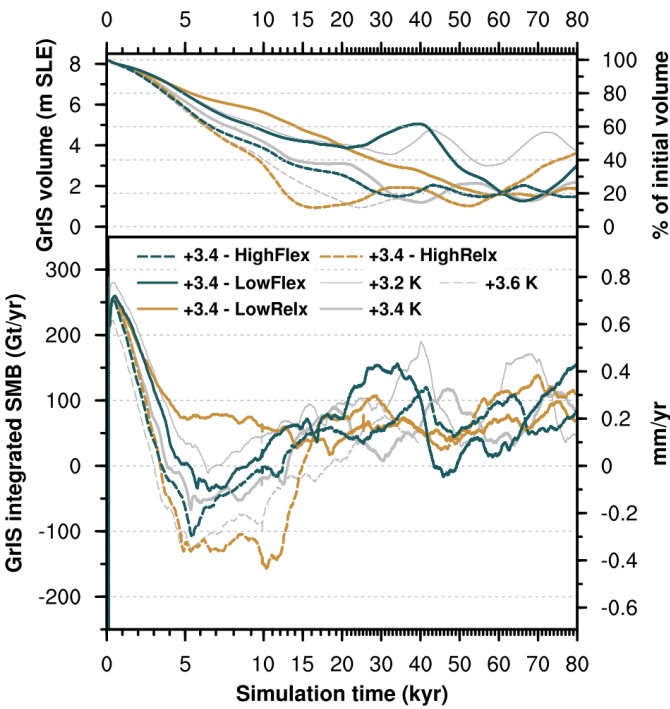

**Figure 5.** Timeseries of (upper panel) GrIS volume and (bottom panel) integrated surface mass balance for selected sensitivity simulations with different GIA parameters. In the legend, LowRelx/HighRelx = Low/High values of the characteristic mantle relaxation time; LowFlex/HighFlex = Low/High values of the flexural rigidity of the lithosphere. Values are shown in Table A2. Note that simulation time in the x-axis is irregular (minor tickmarks every 1000 years).

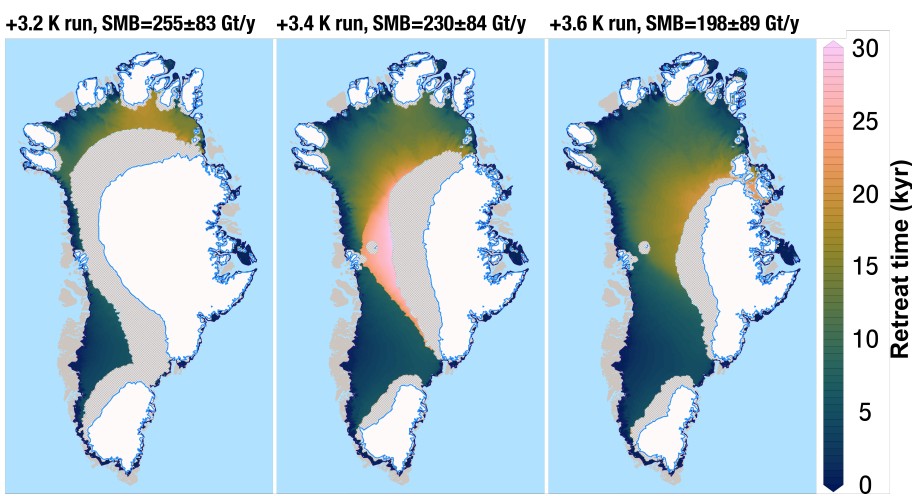

**Figure 6.** Map of GrIS retreat time for the (left) +3.2 K run, (center) +3.4 K run, (right) +3.6 K run. Ice sheet areas showing GIA-induced margin oscillations are shown in grey, whereas areas continuously ice-covered throughout each run are filled in white.

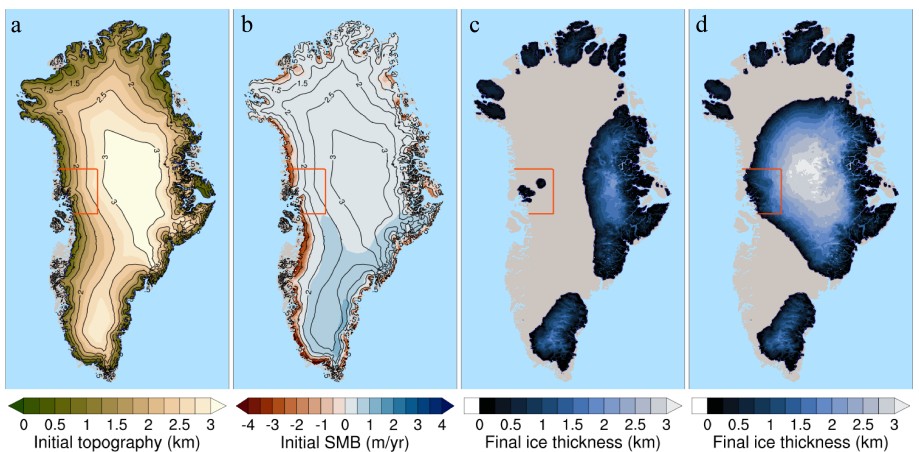

**Figure 7.** Map of (a) GrIS surface topography at the beginning of our simulations, (b) GrIS initial SMB forcing for the +3.4 K run, (c) GrIS final volume for the +3.4 K run, (d) GrIS final volume for the +3.4 K run with constant SMB (*i.e.*, no SMB-elevation feedback) in the central western coastal region (*i.e.*, inside the red box shown in each panel).

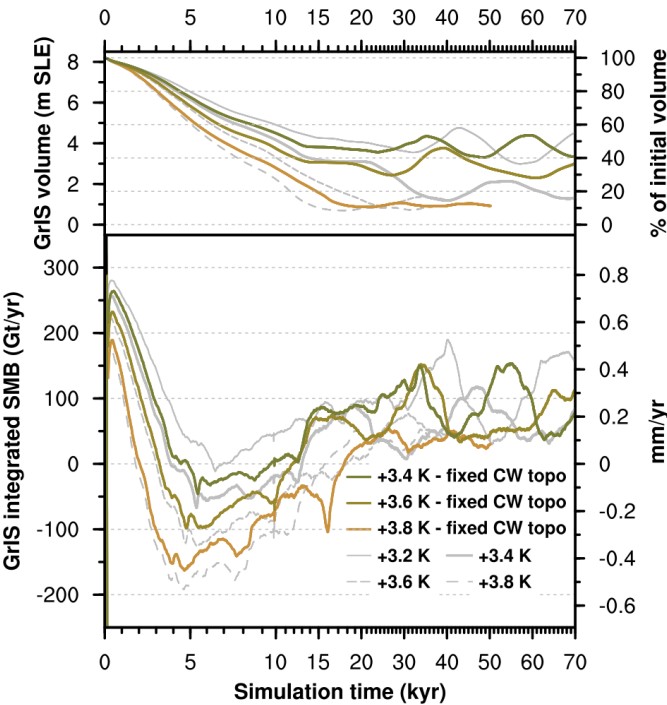

**Figure 8.** Timeseries of (upper panel) GrIS volume and (bottom panel) integrated surface mass balance for selected sensitivity simulations with constant SMB (*i.e.*, no SMB-elevation feedback) in the central western coastal region (*i.e.*, inside the red box in Fig. 7). Sensitivity simulations are labelled with "fixed CW topo". Note that simulation time in the x-axis is irregular (minor tickmarks every 1000 years).

| $\Delta$GMT | $\Delta$GrISMT | Initial SMB forcing | Initial $\Delta$ SMB | Final volume | Minimum volume time |
|---|---|---|---|---|---|
| K | K | avg±stddv, Gt/yr | Gt/yr | min/avg/max, m SLE | kyr |
| **Low melt** | | | | | |
| +2.5 | +2.0 | 361±80 | -230 | 6.66 / **6.79** / 6.91 | 11 |
| +2.8 | +2.2 | 317±97 | -274 | 5.97 / **6.19** / 6.35 | 14 |
| **Medium melt** | | | | | |
| +3.0 | +2.7 | 286±94 | -305 | 4.17 / **4.54** / 4.79 | 35 |
| +3.2 | +2.9 | 255±83 | -336 | 2.99 / **3.95** / 4.76 | 31 |
| **Complete melt** | | | | | |
| +3.4 | +3.2 | 230±84 | -361 | 1.19 / **1.72** / 2.21 | 40 |
| +3.6 | +3.7 | 198±89 | -392 | 0.93 / **1.21** / 1.37 | 25 |
| +3.8 | +4.0 | 156±122 | -435 | 0.70 / **0.82** / 0.95 | 17 |
| +4.0 | +4.2 | 84±158 | -507 | 0.43 / **0.42** / 0.42 | 12 |
| +4.5 | +5.0 | -103±194 | -695 | 0.18 / **0.19** / 0.19 | 8 |
| +5.2 | +5.9 | -387±179 | -978 | 0.07 / **0.08** / 0.09 | 5 |

**Table 1.** First four columns: average values from each of the 23 years-long intervals in the two-way coupled BG-1pct run used to force our simulations (transient global mean temperature anomaly to pre-industrial, transient GrIS temperature anomaly to pre-industrial, GrIS integrated initial SMB forcing, GrIS integrated initial SMB anomaly to pre-industrial). In the last two columns, final GrIS volume (minimum/average/maximum, which are calculated within quasi-periodic oscillations) and time needed to reach minimum GrIS volume in our simulations (*i.e.*, time at which the GrIS retreat stops and quasi-periodic oscillations start.

| Runs | Pre-ind. | Contemporary (1995-2014) | Mid-century (2031-2050) | End-of-century (2080-2099) |
|---|---|---|---|---|
| **CESM2/CISM2** | | | | |
| Pre-ind. | 585±85 Gt/yr | | | |
| Hist. | | 571±80 Gt/yr | | |
| SSP5-8.5 | | | 359±84 Gt/yr | -511±283 Gt/yr |
| **CESM2-only** | | | | |
| Hist. | | 390±28 Gt/yr | | |
| SSP1-2.6 | | | 252±65 Gt/yr | 88±97 Gt/yr |
| SSP2-4.5 | | | 267±58 Gt/yr | 21±80 Gt/yr |
| SSP3-7.0 | | | 227±76 Gt/yr | -269±106 Gt/yr |
| SSP5-8.5 | | | 192±90 Gt/yr | -906±307 Gt/yr |

**Table 2.** Comparison of GrIS integrated SMB (avg±stddev) for two-way coupled CESM2/CISM2 simulations and CESM2-only simulations for pre-industrial, historical and available SSP scenarios contributing to CMIP6. Note that in the CESM2-only runs the SMB is calculated and integrated over the prescribed, present-day observed GrIS elevation. Data from supporting information in Muntjewerf et al. (2020a).

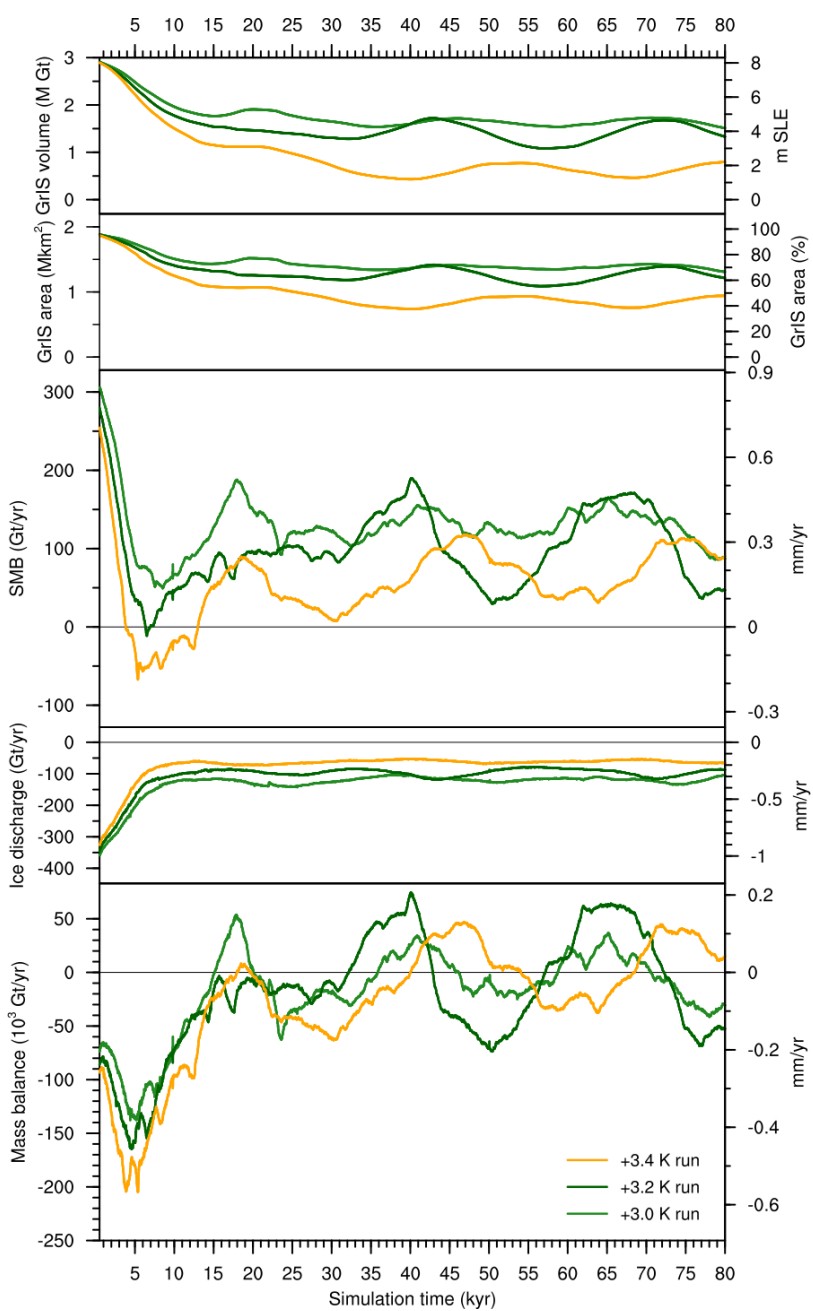

**Figure A1.** Timeseries of GrIS volume, area, surface mass balance, ice discharge and mass balance for selected simulations extended until 80 kyr, to show the quasi-periodicity of GIA-induced oscillations.

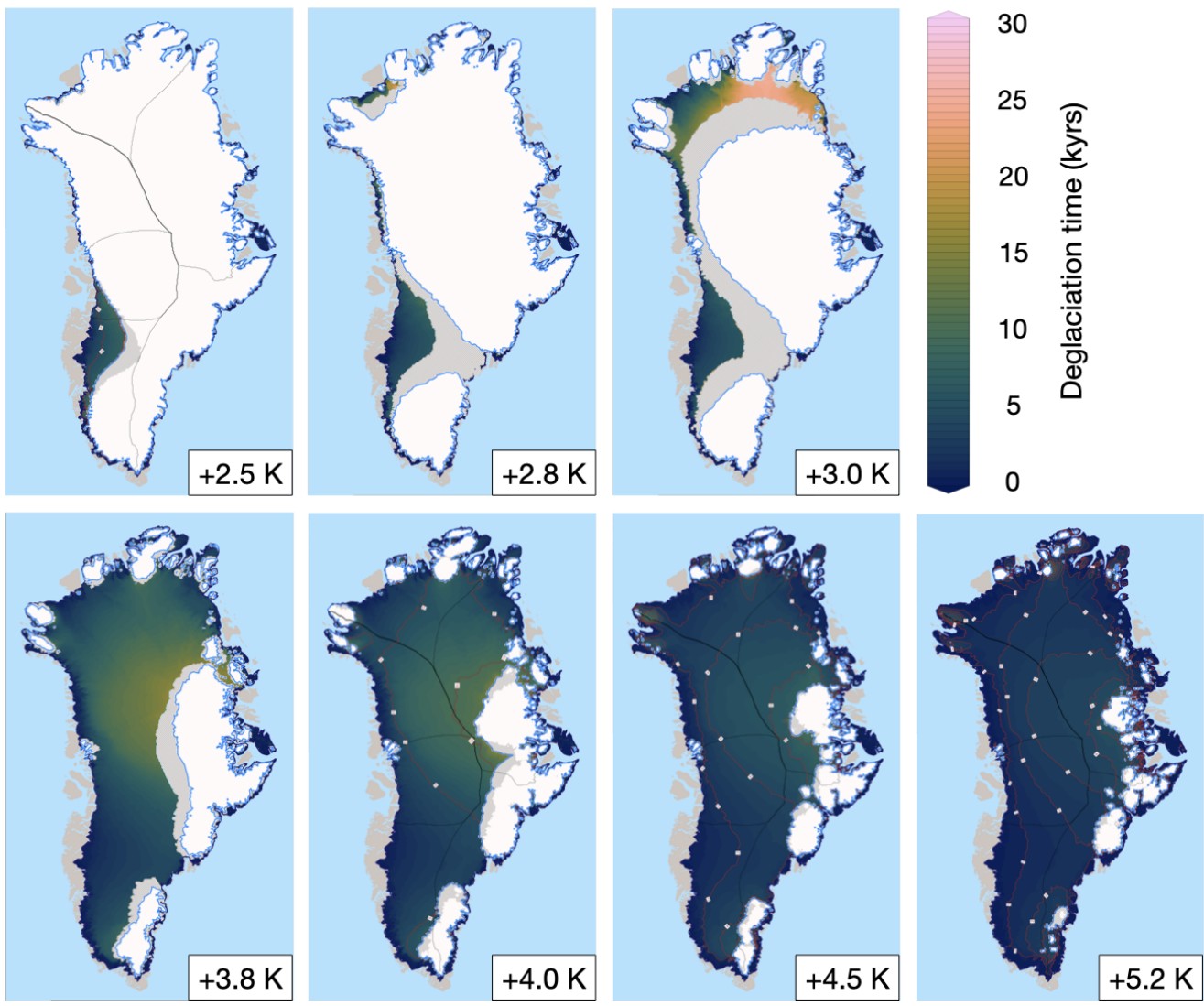

**Figure A2.** Map of GrIS retreat time for the simulations not shown in Fig. 6. Ice sheet areas showing GIA-induced margin oscillations are shown in grey, whereas areas continuously ice-covered throughout each run are filled in white.

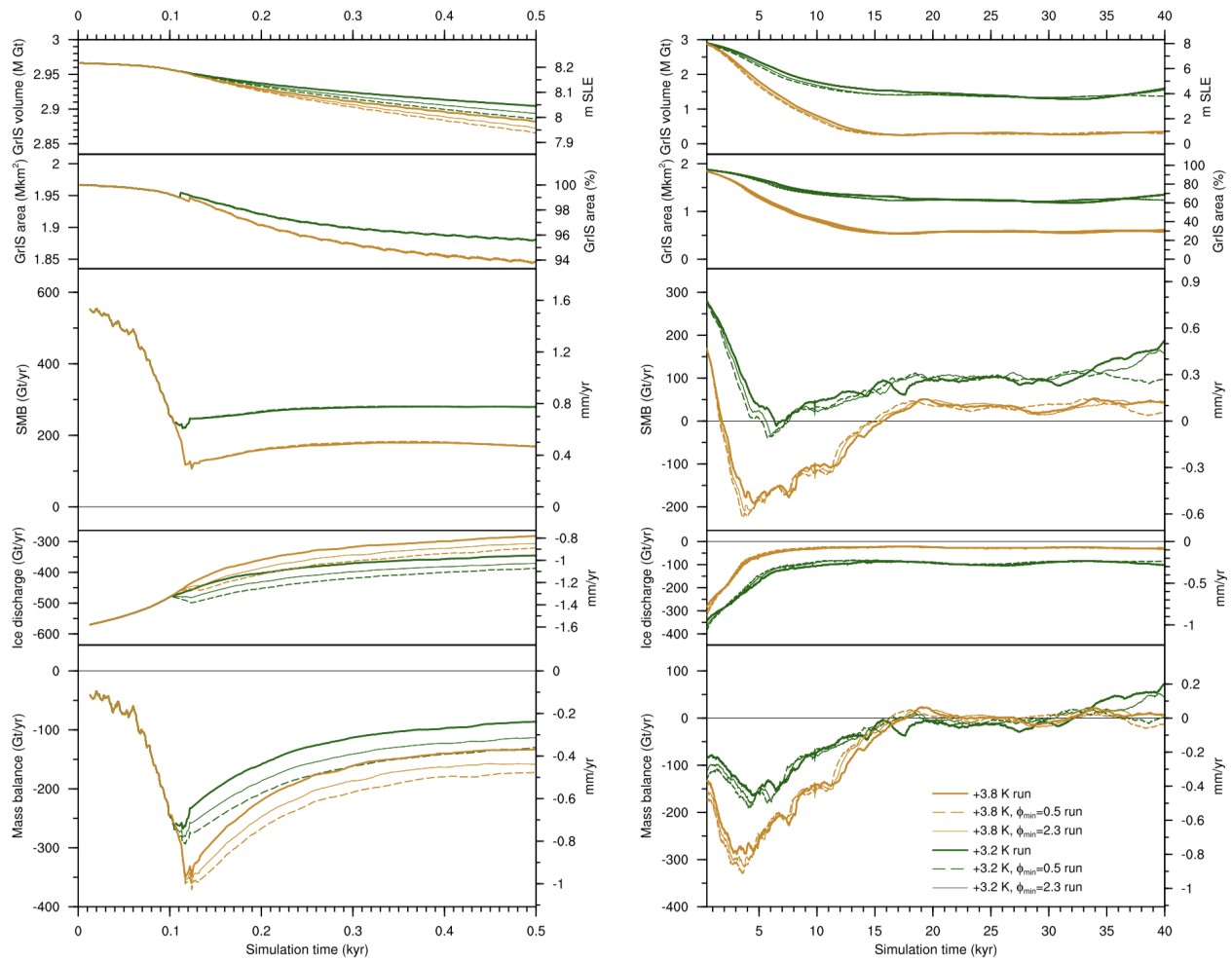

**Figure A3.** Timeseries of GrIS volume, area, surface mass balance, ice discharge and mass balance for selected sensitivity simulations with increased ice discharge through reduced minimum friction angle in pseudo-plastic sliding law (solid thick line: $\phi_{min}$=5, solid thin line: $\phi_{min}$=2.3, dashed line: $\phi_{min}$=0.5. On the left, values between in the first 500 years are shown. On the right, we show values between 500 and 40,000 years.

| Parameter | Short description | Value |
|---|---|---|
| $F_g$ | Constant geothermal heat flux | 0.05 W/m$^2$ |
| $\phi_{\max}$ | Maximum friction angle in pseudo-plastic sliding law, for B $\geq$ B$_{\max}$ | 40° |
| $\phi_{\min}$ | Minimum friction angle in pseudo-plastic sliding law, for B $\leq$ B$_{\max}$ | 5 [2.3, 0.5] ° |
| B$_{\max}$ | Bed elevation above which $\phi = \phi_{\max}$ | 700 m |
| B$_{\min}$ | Bed elevation below which $\phi = \phi_{\min}$ | -300 m |
| q | Exponent for pseudo-plastic sliding law | 0.5 |
| u$_0$ | Threshold velocity for pseudo-plastic sliding law | 100 m/yr |
| $l_{\mathrm{p}}$ | Lithosphere update period | 100 yr |
| $\lambda_f$ | Flexural rigidity of the lithosphere | $0.24 \cdot 10^{25}$ [ $0.24 \cdot 10^{24}$, $0.24 \cdot 10^{26}$] Pa m$^3$ |
| $\tau_r$ | Characteristic mantle relaxation time | 3000 [1000, 6000] yr |

**Table A1.** Model parameters used in the main CISM2 simulations presented in this study. Parameter values tested in sensitivity simulations are indicated in brackets.)

| Simulations | Forcing years | SMB (avg $\pm$ stddv) | $\Delta$SMB |
|---|---|---|---|
| +2.5 K | 74-96 | 361 $\pm$ 80 Gt/yr | -274 Gt/yr |
| +2.8 K | 81-103 | 317 $\pm$ 97 Gt/yr | -305 Gt/yr |
| +3.0 K | 86-108 | 286 $\pm$ 94 Gt/yr | -336 Gt/yr |
| +3.2 K | 90-112 | 255 $\pm$ 83 Gt/yr | -361 Gt/yr |
| +3.4 K | 95-117 | 230 $\pm$ 84 Gt/yr | -392 Gt/yr |
| +3.6 K | 99-101 | 198 $\pm$ 89 Gt/yr | -435 Gt/yr |
| +3.8 K | 103-125 | 156 $\pm$ 122 Gt/yr | -435 Gt/yr |
| +4.0 K | 107-129 | 84 $\pm$ 158 Gt/yr | -507 Gt/yr |
| +4.5 K | 116-138 | -103 $\pm$ 194 Gt/yr | -695 Gt/yr |
| +5.2 K | 131-150 | -387 $\pm$ 179 Gt/yr | -978 Gt/yr |

**Table A2.** Values from each of the 23 years-long intervals in the two-way coupled BG-1pct run used to force our simulations (transient global mean anomaly to pre-industrial, initial and final forcing years, GrIS integrated SMB, GrIS integrated SMB anomaly to pre-industrial).