# Peer review of "A topographically-controlled tipping point for complete Greenland ice sheet melt"

_The Cryosphere, 2023_

## Author Comment (AC1)

We thank the three anonymous referees for their valuable comments and constructive criticism.

All the referees' minor comments (including Figures, Tables, and typos) will be incorporated in the revised version of the manuscript. We will provide tracked changes and a detailed description of how, and where, each minor comment is addressed in the revised manuscript.

Below, we explain how the main (major) comments will be addressed in the revised manuscript:

1) **Clearly defining the meaning of tipping point** (raised by all referees), and more in general, **avoiding too vague terminology in the text** (raised by Referee #1);

In the revised version of the manuscript the meaning of tipping point will be clearly defined early in the text. To directly answer the referees: in this study we refer to the following definition of tipping point from Lenton, Early warning of climate tipping points, Nature Climate Change, 2011: *"A climate 'tipping point' occurs when a small change in forcing triggers a strongly nonlinear response in the internal dynamics of part of the climate system, qualitatively changing its future state".* This is exactly what we find in our study: a minor change in the initial Surface Mass Balance forcing (SMB: from 255 Gt/yr to 230 Gt/yr) and global mean temperature (from 3.2 K to 3.4 K) triggers an abrupt nonlinear change in the Greenland ice sheet equilibrium volume (from 50% mass to nearly complete deglaciation). Our study does not aim to address either the multistability or the reversibility of the Greenland ice sheet: we will add explicit clarification in the manuscript. For instance, in the introduction:

*"Although the concept of a 'tipping point' is often associated with irreversibility, hysteresis, and self-perpetuating mechanisms, it is important to clarify that the primary focus of our study is not to investigate the multistability or the reversibility of the Greenland Ice Sheet (GrIS). Instead, we are primarily concerned with understanding the potential surface mass balance (SMB) threshold for complete melt of the GrIS, the processes that control this threshold, and whether it exhibits characteristics commonly associated with tipping points, such as sensitivity to external forcings.",*

and in the discussion section 4.2 where we discuss the implications:

*"In this discussion section, we examine the behavior of the Greenland Ice Sheet (GrIS) with a specific focus on the potential tipping point associated with the surface mass balance (SMB) threshold for complete melt. While we discuss the GrIS's responses to sustained melt, the non-linear and abrupt nature of those responses and the sensitivity to external factors, it is important to reiterate here that our primary objective is to explore the SMB threshold itself, rather than the aspects of multistability or reversibility often associated with tipping points. "*

As for other too vague terminology ("runaway", "stabilising", "topographically-controlled tipping point", "pinning point"), changes will be made to define specific terminology upfront or rephrase more clearly.

2) **Estimating uncertainties to test the robustness of the presented result** (raised by Referee #1);

To explore uncertainties related to the GIA parametrization, we have repeated two key simulations (before and after tipping point, i.e. +3.2 K and +3.4 K, respectively) while varying two model parameters (i) the viscoelastic relaxation time of the mantle (default value: 3000 years; values explored 1000, 6000, 10000), (ii) the lithosphere flexural rigidity (default value: 0.24e25; values explored 0.24e24, 0.24e26). The choice of model parameters and their values was guided by results presented in Zweck and Huybrechts, Modeling of the northern hemisphere ice sheets during the last glacial cycle and glaciological sensitivity, Journal of Geophysical Research, 2005. From a first-order analysis of the new set of runs (ice volume timeseries) our results seem robust: the Greenland ice sheet loses more than 80% of its mass for

a +3.4 K forcing independently from the values of the mantle viscoelastic relaxation time and lithosphere flexural rigidity. A more detailed analysis of this set of experiments will be included in the new version of this manuscript, drawing conclusions on how robust the SMB threshold with respect to the GIA parametrization is.

Referee #1 has also asked us to assess the robustness of our results for different values of the lapse-rate air temperature correction. Unfortunately, this is not technically feasible. The SMB forcing at multiple Elevation Classes used in our study has been previously calculated (using a fixed lapse rate air temperature correction of -6 K/km) in the fully coupled CESM2/CISM2 1pctCO2 simulation described in Muntjewerf et al., [Accelerated Greenland Ice Sheet Mass Loss Under High Greenhouse Gas Forcing as Simulated by the Coupled CESM2.1-CISM2.1](), JAMES, 2020. Testing the sensitivity of the SMB threshold found here to the temperature lapse-rate correction would require coupled land/ice sheet model (CLM5-CISM2) simulations forced with atmospheric forcing data (see "CLM5.0 User's Guide, Section 1.5.7"). This model setup is in theory feasible but has never been tested before. Moreover, 3-hourly (!) atmospheric forcing data needed to force such land/ice sheet simulations are not available, as they were not saved from the fully coupled 1pctCO2 CESM2/CISM2 simulation (Muntjewerf et al 2020) to avoid excessive output storage. Therefore, producing atmospheric forcing data would require first running again fully coupled CESM2/CISM2 segments. Overall, running additional simulations suggested by Referee #1 to test the impact of the lapse would require an amount of model development and computing time beyond those available for this study. The impact of different lapse-rate correction values on the CLM prognostic melt energy and SMB gradients over the Greenland ice sheet using the Elevation Classes method has been discussed in detail in Sellevold et al., [Surface mass balance downscaling through elevation classes in an Earth system model: application to the Greenland ice sheet](), The Cryosphere, 2019. In the revised version of the manuscript, we will include this reference, and an explanation of why different values of the temperature lapse-rate correction cannot be explored.

3) **Proving our hypothesis that the topography is a major control on the Greenland ice sheet tipping point** (raised by Referee #1);

We thank Referee #1 for suggesting an interesting approach to strengthening our case, that is: "*If the authors wanted to prove this statement, they could for example prevent this region from deglaciating in the strong warming simulations (by not changing the SMB there for example) and showing that this means also the rest of the ice sheet does not vanish*". We followed this suggestion by repeating 'high-melt' simulations +3.4 K, +3.6 K, and +3.8 K while inhibiting the SMB-height feedback around the high-topography region in the Central West (red region in Figure 1 below). This means that in the highlighted region the SMB is held constant, as suggested by Referee #1. Keeping the SMB fixed in this region prevents the Greenland ice sheet complete mass loss in the 3.4 K and 3.6 K runs, with a final ice sheet volume of ~50% and ~40% of the preindustrial value. This proves that the surface topography in the central West exerts a major control on the ice sheet tipping point. In the 3.8 K run, the SMB forcing is sufficiently high for the ice sheet to detach from the central western fixed-SMB region and then rapidly retreat all the way to the East. The revised version of the manuscript will include a thorough description of these new experiments, with additional figures showing the bedrock and surface topography in the central West.

[Figure]

Figure 1: the red area indicates the region where the SMB-height feedback is inhibited (and as such, the SMB is held constant).

---

## Author Response (AR1)

We thank the three anonymous referees for their valuable comments and constructive criticism.

All the referees' minor comments (including Figures, Tables, and typos) have been incorporated in the revised version of the manuscript (in the section 'Minor comments' we describe how, and where, each minor comment is addressed in the revised manuscript). Below, we explain how the main (major) comments have been addressed in the revised manuscript. In short, in the revised manuscript:

- We clearly define what we mean by tipping point, and we removed too vague terminology.
- We present and discuss uncertainties related to GIA model parameters; we discuss why we cannot test different lapse rate values, and we discuss the implications.
- We performed additional test to strengthen our case that the topography in the central West is a key control on the Greenland ice sheet threshold behavior.

**Major comments:**

1) **Clearly defining the meaning of tipping point** (raised by all referees), and more in general, **avoiding too vague terminology in the text** (raised by Referee #1);

In the revised version of the manuscript the meaning of tipping point is clearly defined early in the text. In this study, we refer to the following definition of tipping point from Lenton, Early warning of climate tipping points, Nature Climate Change, 2011: *"A climate 'tipping point' occurs when a small change in forcing triggers a strongly nonlinear response in the internal dynamics of part of the climate system, qualitatively changing its future state".* This is exactly what we find in our study: a minor change in the initial Surface Mass Balance forcing (SMB: from 255 Gt/yr to 230 Gt/yr) and global mean temperature (from 3.2 K to 3.4 K) triggers an abrupt nonlinear change in the Greenland ice sheet equilibrium volume (from 50% mass to nearly complete deglaciation). Our study does not aim to address either the multistability or the reversibility of the Greenland ice sheet. We have included explicit clarification throughout the revised manuscript (L43, L86, L289). Other terminology indicated as "too vague" has been entirely removed ("runaway", "stabilising", "topographically-controlled pinning point").

2) **Estimating uncertainties to test the robustness of the presented result** (raised by Referee #1);

To explore uncertainties related to the GIA parametrization, we have repeated two key simulations (before and after tipping point, i.e. +3.2 K and +3.4 K, respectively) while varying two model parameters (i) the viscoelastic relaxation time of the mantle (default value: 3000 years; values explored 1000, 6000), (ii) the lithosphere flexural rigidity (default value: 0.24e25; values explored 0.24e24, 0.24e26). The choice of model parameters and their values was guided by results presented in Zweck and Huybrechts, Modeling of the northern hemisphere ice sheets during the last glacial cycle and glaciological sensitivity, Journal of Geophysical Research, 2005. Results are shown in the revised manuscript (in particular in Section 3.3) and are discussed in subection 4.3 "Conclusion and future research". Referee #1 has also asked us to assess the robustness of our results for different values of the lapse-rate air temperature correction. Unfortunately, this is not technically feasible. The SMB forcing at multiple Elevation Classes used in our study has been previously calculated (using a fixed lapse rate air temperature correction of -6 K/km) in the fully coupled CESM2/CISM2 1pctCO2 simulation described in Muntjewerf et al., Accelerated Greenland Ice Sheet Mass Loss Under High Greenhouse Gas Forcing as Simulated by the Coupled CESM2.1-CISM2.1, JAMES, 2020. Testing the sensitivity of the SMB threshold found here to the temperature lapse-rate correction would require coupled land/ice sheet model (CLM5-CISM2) simulations forced with atmospheric forcing data (see "CLM5.0 User's Guide, Section 1.5.7"). This model setup is in theory feasible

but has never been tested before. Moreover, 3-hourly (!) atmospheric forcing data needed to force such land/ice sheet simulations are not available, as they were not saved from the fully coupled 1pctCO2 CESM2/CISM2 simulation to avoid excessive output storage. Therefore, producing atmospheric forcing data would require first running again fully coupled CESM2/CISM2 segments. Overall, running additional simulations suggested by Referee #1 to test the impact of the lapse would require an amount of model development and computing time beyond those available for this study. The impact of different lapse-rate correction values on the CLM prognostic melt energy and SMB gradients over the Greenland ice sheet using the Elevation Classes method has been discussed in detail in Sellevold et al., Surface mass balance downscaling through elevation classes in an Earth system model: application to the Greenland ice sheet, The Cryosphere, 2019. In the revised version of the manuscript, an explanation of why different values of the temperature lapse-rate correction cannot be explored and a discussion of its implications are included in subsection 4.3.

3) **Proving our hypothesis that the topography is a major control on the Greenland ice sheet tipping point** (raised by Referee #1);

We thank Referee #1 for suggesting an interesting approach to strengthening our case, that is: "*If the authors wanted to prove this statement, they could for example prevent this region from deglaciating in the strong warming simulations (by not changing the SMB there for example) and showing that this means also the rest of the ice sheet does not vanish*". We followed this suggestion by repeating 'high-melt' simulations +3.4 K, +3.6 K, and +3.8 K while inhibiting the SMB-height feedback around the high-topography region in the Central West (red region in Figure 1 below). This means that in the highlighted region the SMB is held constant, as suggested by Referee #1. Keeping the SMB fixed in this region prevents the Greenland ice sheet complete mass loss in the 3.4 K and 3.6 K runs, with a final ice sheet volume of ~50% and ~40% of the preindustrial value. This proves that the surface topography in the central West exerts a major control on the ice sheet tipping point. In the 3.8 K run, the SMB forcing is sufficiently high for the ice sheet to detach from the central western fixed-SMB region and then rapidly retreat all the way to the East. The revised version of the manuscript includes a thorough description of these new experiments, with additional figures showing the bedrock and surface topography in the central West.

**Minor comments:**

**Abstract**

R1: line 12: "highly nonlinear" → "nonlinear" as highly is not quantified.
Addressed, we removed 'highly'.
R1: line 14 "runaway retreat", runaway means that the feedback gain of this process if larger than 1, I don't think that this has been shown yet. Rather, you can say "self-reinforcing", "self-sustained".
Addressed, we removed 'runaway retreat' and we rephrased the sentence.
R1: line 14: if GIA only delays deglaciation, it does not "stabilise" the margin in the sense of stability analysis. It only affects the timescales.
Addressed, we removed 'stabilise' and we rephrased the sentence.
R1: line 14-16: this sentence is not clear, reformulate.
Addressed, we rephrased the sentence.
R1: line 16-18: you do not show that this region is the trigger for tipping.

Corrected, see response to major comment number 3.

R1: line 19: what do you mean with "stabilising effect"?

We rephrased the last sentence to make it more clear.

**Introduction**

R1: in general the introduction is written as if you are intending to provide an estimate for a tipping point, however, this is not done in this manuscript (see main comments).

Please see response to major comment number 1.

R1: line 24-25: Add a citation for "while until the late 1990s ice discharge has been the main source of ice loss".

Added citation Shepherd et al. 2012, Science.

R1: GIA and it's effect on GIS stability (papers like Zeitz et al.,2022) is directly relevant to this study and should be discussed in the introduction.

We agree, thanks for pointing this out. We included a few sentences in the introduction discussing GIA and its impacts (there is a new paragraph focusing on feedbacks for long-term GrIS change, lines 42-63 in the revised manuscript).

R2: In the abstract and L65 the phrase occurs "and we explore the processes controlling the nature of this threshold, [..]". I would suggest you state upfront which processes you find to control the threshold instead of leaving the reader in suspense.

Done, thanks for the suggestion.

R2: Please define tipping point carefully as it is usually taken to mean non-reversible state transitions once a threshold (of a control parameter) is reached. It is not sufficiently clear if you take it to mean this, but if so, please give a short account of how this can occur when considering the SMB-height feedback with GIA.

Done,

R2: L66: Not sufficiently clear what "SMB levels" refers to at this point. Please help the reader a bit more here.

Done, replace it with 'SMB climatologies' (also in the abstract).

R3: The first paragraph is really long, can you divide it into several smaller ones? The SMB-height feedback appears in the middle of the paragraph. It is the main focus (along with the competing effect of the GIA) of your work but it doesn't really stand out in the text.

Done, thanks. We split the introduction in three main paragraphs. One paragraph is now dedicated to SMB-height feedback, GIA, and other feedback mechanisms (lines 42-63 in the revised manuscript).

R3: Starting a new paragraph here would be a good idea. You could also put a bit more emphasis on this process to attract the reader's attention. This could be done by putting your definition of tipping point just before (if you can justify the use of the words tipping point) or by rewritting line 36 to 38. In particular, line 36 says that the SMB-height feedback is "another important source for future GrIS mass loss". This is absolutely true but, without developing a bit more on the feedback, you make it sound like you're just discussing another source of ice loss instead of the main point of your work.

See above. We also included a mention to GrIS tipping point at the beginning of the paragraph (line 42 in the revised manuscript): ss ice sheets are known to respond to surface changes over centuries and millennia rather than decades, an extremely important question is whether over longer timescales ongoing and projected 21st century SMB changes could eventually lead to complete GrIS melt, possibly through self-perpetuating mechanisms triggering a non-linear and abrupt response commonly associated with tipping points.

R3: The competition between melt and GIA is also an important point in your work but you only mention at the end of your introduction that CISM2 takes into account the effects of GIA. Please add a sentence about the fact that it will counterbalance the effect of the melt and influence how melt is evolving throughout the simulations.

Done, thanks. Specifically at lines 55-62 in the revised manuscript.

R3: Line 1: currently storing

Corrected, thanks.

R3: Line 26: commas around "as of today"

Corrected, thanks.

R3: Line 45: no comma in "either explicitly, or parametrized"

Corrected, thanks.

R3: Line 49: commas around "during the last interglacial period (LIG…)"

Corrected, thanks.

R3: Line 51: suggests

Corrected, thanks.

R3: lines 55-57: This is a really long sentence. Cutting it after warmer climate and starting a new one with However improves readability

Corrected, thanks.

**Methods**

R1: A section on the model initialisation is missing. It is pointed to previous studies, but it would be useful to have a quick summary how you initialise and optimise model parameters here.

We included a section about model initialization in lines 152-163, thanks for pointing this out.

R1: lines 97 and following: does the downscaling mean that the SMB that the ice sheet model sees is not equivalent in mass to the fluxes in CESM2?

We included some text in the revised manuscript addressing this, see lines 137-138: "Remapping onto the CISM2 grid is followed by a normalization step that guarantees conservation of SMB fluxes in both the accumulation and ablation zones". For more details, see Section 6.4.1 in the CESM Land Ice Documentation and User Guide.

R1: line 129: How long does it take to reach the stable ice sheet configuration? What is your criterion for "stability"? Looking at Figure 2, it appears that quite a few runs are still changing (2.5 to 3.8K runs).

Since explaining this would require discussing already the results, we replaced the text with "In each simulation, we cycle the 23 years-long, multiple-ECs SMB forcing for 80 kyrs (or less if the ice sheet disappears earlier)."

R1: How are the sensitivity runs done? You mention different till friction angle and switching off bedrock uplift – are new equilibrium initial states created for these conditions? If not, do you show the results in FigA2 relative to a control run?

We expanded the paragraph describing setup and scope of the sensitivity tests (including new ones, see major comment number 2.

R3: Lines 115 - 127:  I'm not sure I understand things correctly here. From section 2.3 I understand that your work starts from the already existing CESM2/CISM2 simulation of Muntjewerf et al., 2020b and that you use different points in time from that simulation to force CISM2 in your simulation. In line 115 you also mention that you run CISM2 within the CESM2 architecture. Do you mean that you completely bypass the atmospheric and land modules and directly force CISM2 with the SMB from the already existing

Muntjewerf simulation? If yes, is it the novel forcing method that you mention at the beginning of the section. If not, do you mean you've been running the same configuration as Muntjewerf et al. and used snapshots of it as initial conditions and that the SMB was calculated during your runs?

"Do you mean that you completely bypass the atmospheric and land modules and directly force CISM2 with the SMB from the already existing Muntjewerf simulation?": yes, correct. We reorganized the text and rephrased this section to (hopefully) make it more clear to the reader.

R3: You also mention later (in the discussion) that CISM2 is not bi-directionally coupled to CESM2. This is a shortcoming of your study as it makes the comparison with Gregory et al. 2020 a bit more difficult since, as you also say in the discussion, you don't take into account the effects of a changing ice sheet geometry on the local climate. Please add a sentence after line 120 mentioning the absence of coupling the other way, that it doesn't take into account the effect of the ice sheet on the climate and a justification of why you're doing it that way. I suppose that the only reason, which is entirely justifiable, is that the computing resources that would be needed to do that would be almost astronomical.

Done. That's right, this approach allows to explore longer timescales while using the CESM2 coupling functionality, but not its physics - computational costs would indeed be nonsensical.

R3: Line 81: model instead of Model

Corrected.

R3: Line 103: -2°C and 0°C instead of K

Right, corrected.

R3: line 107: missing space after 1°

Corrected.

R3: line126: SMBs instead of SMB

We rephrased this section, but thanks for spotting this.

R3: line127: "results from a compromise" instead of "results as a compromise"

Corrected, thanks.

**Results**

R1: The concept of a "topographic pinning point" is not clear to me. The results are meant to show that a region in west Greenland is such a "pinning point", and if it is "un-pinned" the GrIS retreats in all other regions as well. However, the results show only a correlation between this region being part of the maximum ice sheet extent during oscillations and regions in Greenland being glaciated. And it disappears first. This does not mean that this specific region has any control over other regions being glaciated further away. There is no physical process named through which this pinning point is "stabilising" the other regions of the ice sheet.

Please see response to major comment number 3.

R1: line 139 "exhibits"

Corrected, thanks.

R1: line 139 how do you quantify "highly nonlinear"?

Addressed, we removed 'highly'.

R1: line 146: please clarify "low GrIS melt is achieved for a decrease in SMB not exceeding 50% of the equilibrium pre-industrial SMB". This sentence is quite complicated. And why is the pre-industrial SMB an equilibrium SMB?

We rephrased this sentence. Also, the correct number here was 40%, thanks for helping us to spot that. "equilibrium SMB" because the coupled CESM2/CISM2 pre-industrial simulation is run until equilibrium (we clarified that in the manuscript).

R1: line 177: what do you mean with "more stable"?

The use of 'more' was uncalled for, we removed it. Thanks for spotting that.

R1: line 193: discharge is not included in your experiments except for a fixed

The sentence here was not complete, but we tried to interpret what the reviewer was asking: how can you test increased discharge if there is no ice-ocean interaction? We rephrased this part, explaining how we trigger an increase in ice discharge (through higher sliding, by changing sliding parameters in the pseudo-plastic sliding law). We described better why we claim that in our simulation the SMB signal is dominating the long-term GrIS response.

R1: line 197: please specify where this region is in the figures, e.g., show it in the figures.

Done, thanks (Fig. 7).

R1: line 197: what is a topographic pinning point?

We opted for removing this (and other) vague terminology throughout the text (see also response to major comment 1).

R1: line 201: what do you mean with "runaway"?

We opted for removing this (and other) vague terminology throughout the text (see also response to major comment 1).

R1: line 203: How is this tipping point behaviour? You do not show irreversibility.

See response to major comment 1.

R2: Please define the SMB threshold clearly since you take it to mean Gt/yr in some places, but in other places refer to the GMT increase of some experiment? E.g. in L280-282 you do this nicely, but would be helpful to have a similar clarification earlier.

We rephrased the whole subsection to make it clearer, as suggested.

R2: L139: "exhibits"

Corrected, thanks.

R3: Line 145-147: can you add the equilibrium pre-industrial value in the text to make it easier to compare? Like you did at the beginning of the discussion section.

That's right, done it. We also realized we provided a different value than in Muntjewerf et al. 2020 (591±83 instead of 585±85), it is now corrected.

R3: Line 160: "timescales for ice loss increase sharply…" Do you mean the timescale needed to reach the final volume/equilibrium?

Yes, we corrected the text in 'Timescale needed to reach ice sheet deglaciation or stabilization increases sharply'. We did not want to use the word equilibrium as we would need to introduce too early the concept of quasi-equilibrium and quasi-periodic oscillations (which are hard to define before talking about specific processes for mass loss/mass gain).

R3: Line 139: exhibits

Corrected, thanks.

R3: Line 140: misplaced space in "Table1, the"

Thanks for spotting this.

R3: Line 163: "longest" instead of "highest"

Corrected, thanks.

R3: Line 174: comma after "consequently" (or no comma before)

Corrected, thanks.

R3: Line 175: missing video reference (xxx)

Corrected, thanks.

R3: Line 177: Fig. 2 - missing space

Corrected, thanks for spotting this.

R3: Line 180: "from 3 to 12%" instead of "between 3% and 12%"

Corrected, thanks.

R3: Line 183: oscillations

Corrected, thanks.

Line 197: acts

We rephrased this part, but thanks for spotting this.

R3: Line 200: permanently loses

Corrected, thanks.

R3: Line 205: "in around 15kyrs or less" or "around 15kyrs or sooner"

Corrected, thanks.

**Discussion**

R1: line 219: it might be worth giving the numbers here as they diverge quite a lot

Done, thanks.

R1: line 241: remove "highly"

Done, thanks.

R2: L208: I think it would be helpful to expand on what is meant by "Even though our simulations are not meant to represent realistic scenarios" since you then later go on to compare with, what I would regard, attempts in the literature of make realistic model experiments.

We rephrased this initial paragraph, and we decided to remove this sentence, as an explanation of why our comparison with future projections should be considered carefully is given later in the text.

R2: Please consider qualifying early on that "complete ice-sheet melt" also means "complete ice sheet (mass) loss". E.g. L284 you do this excellently, but having this clearly defined early on, too, would be helpful.

Thanks for the suggestion. The first sentence of the result section now reads "In our simulations, the GrIS exhibits a sharp threshold behaviour, with a nonlinear relationship between initial SMB forcing and GrIS melt/mass loss".

R2: Not sufficiently clear which experiments are fully coupled and which are not; e.g. L247 vs. L266-267 and in conclusions.

Modified the text to make this more clear (e.g.,: L315 in the revised manuscript: "In their study, Gregory et al. (2020) used an ISM coupled to a low-resolution atmosphere GCM, and used ECs to downscale the SMB onto the ice sheet model grid. While in our simulations we also use ECs to downscale the SMB forcing onto the CISM2 grid, GrIS changes are not communicated back to CESM2, and as such we do not account for a number of ice-atmosphere feedbacks (e.g., increased cloudiness and snowfall as the ice sheet margin moves towards the interior". Changes have been made also at L337 and in the conclusions).

R3: Line 215: what type of applied corrections do you mean? I thought you meant that the Lofverstrom simulation was not coupled but line 218 tells me this is not the case.

We clarified in the text the meaning of corrections, see L282 in the revised manuscript: "This corresponds to a 60% decrease from the GrIS pre-industrial SMB calculated in a two-way coupled CESM2/CISM2 equilibrium simulation (585±83 Gt/yr, see Table 2 and Lofverstrom et al., 2020). In this pre-industrial two-way coupled ESM/ISM simulation no corrections are applied to the SMB (e.g., SMB masking outside the present-day GrIS extent) and the GrIS is left free to evolve"

R3: Line 246: please add that the model used in Gregory et al. is bi-directionally coupled here since you mention that the version of CESM/CISM you are using is not in the next sentence → "used an ISM bi-directionally coupled to"

Done, thanks.

R3: Line 218: comma before "as a consequence"

Done, thanks.

R3: Line 218: no comma after "SMB"

We rephrased the sentence removing the typo.

R3: lines 249-250: commas around "which … feedback" + lead or leads (increase in precipitation leads to or these processes lead to)?

We rephrased the sentence removing the typo.

R3: Line 258: "agrees well with Zetz et al., 2022 and Plach et al., 2019"

Corrected, thanks.

R3: Line 262: also limited

Corrected, thanks.

R3: Line 286: promotes

Corrected, thanks.

R3: Line 286: which in turn yields

Corrected, thanks.

R3: Line 290: has also been

Corrected, thanks.

R3: Line 297: includes

Corrected, thanks.

R1: line 290: provide the "other studies"

Done, thanks.

R1: lines 291: you do not show that "tipping" depends on the margin – you show that the margin gets lost first.

See response to major comment 3.

R1: line 294: give the previous modelling studies

Done, thanks.

R1: line 297: "includes", typo

Corrected, thanks.

**Tables and Figures**

R1: Table 1: The SMB related to the initial values at the start of the repeat-forcing runs, or? Where do the uncertainties in the final volume come from? Why are there no uncertainties in the Min. vol. Time then?

Yes, correct. We specified that in the revised version of the manuscript ('Initial SMB forcing'). Uncertainties come from the min/max volume oscillations. Min. vol. time is defined as the time minimum

volume is reached before oscillations start (we specified that in the caption in the revised version of the manuscript).

R1: Table 2: What are the CESM2-only runs? It appears to me that they are not relevant to this study here?

These are CESM2 runs with non-active Greenland ice sheet (we included a reference for the data in the revised manuscript). We think they are relevant as they provide some context to the reader of how much SMB is changing in future projections with/without ice sheet coupling.

R1: Figure 2: How are the two axes GrIS integrated SMB (Gt/yr) and mm/yr linked? The GrIS extent changes drastically in the runs, and theoretically, this can cause a change of the integrated SMB while the average SMB remains the same.

They are converted in Sea Level Equivalent.

R1: Figure A5, A6: I think these are never referenced.

That's right, we removed them.

R1: Table A1: does this mean you are using a spatially constant geothermal heat flux?

That's correct.

R3: Figure 1: This is the most important figure of the paper but it's a bit cluttered, which makes it a bit difficult to interpret. A few suggestions to improve the readability: - Increase the space between the 2 subfigures to separate them more clearly - Increase the width of both of them a bit if also possible - Remove "run" in figure 1a legend - Use a slightly less thick font for axis labels

Thanks for the suggestion, we have done what the reviewer suggests.

R3: Table 1: please move the units with the column titles so that the focus can be on the numbers only;

Corrected, thanks.

**Other comments**

R3: You are using a mix of British and American English, please homogenise. I spotted the use of modelling (British spelling), color (American spelling) and both British and American spellings for parametrisation/parametrization. You also use both parametrize and parameterize. It looks like both are correct but could you stick to one spelling?

Corrected, thanks.

Editor: Please ensure that the colour schemes used in your maps and charts allow readers with colour vision deficiencies to correctly interpret your findings. Please check your figures using the Coblis – Color Blindness Simulator (https://www.color-blindness.com/coblis-color-blindness-simulator/) and revise the colour schemes accordingly.

All figures now are done using color schemes from Crameri (2018) to prevent visual distortion of the data and exclusion of readers with colour-vision deficiencies (Crameri et al., 2020).

---

## Referee Report (RR1)

**General statement**

This is the second round of review for this manuscript. Three major comments and a number of more minor comments were raised by the three reviewers. The authors did a great job at addressing the comments and this lead to extensive re-writing of some parts of the manuscript, as well as additional test simulations.

In general, I think that taking into account the reviewers' comments greatly improved an already really interesting manuscript.

In particular, the introduction now addresses a major remark made by all three reviewers about the lack of a definition for tipping point. This section has also been re-written to address a minor comment by reviewer 3. The readability of this section has much improved in the revised version and the point of the paper (existence of a tipping point) and the competing effects (SMB-elevation feedback vs GIA and other negative feedbacks) are now clearly described.

I found that the major comment of reviewer 1 to add a simulation where the SMB was kept constant in the coastal region of central West Greenland (to test the hypothesis that the ice sheet being pinned to a high topography point in this region has a stabilising effect) was a really interesting approach. The new simulations do show that, if the ice sheet is kept pinned on that point, nearly complete loss is prevented at higher levels of warming than if the SMB-elevation feedback is taken into account. The complete disappearance of the «medium melt» category and more abrupt threshold behaviour is to me a further clue of the stabilising effect of this pinning point. Adding a new subsection (3.4) to address this also puts more focus on this phenomenon and justifies the title of the paper more than in the first version.

Finally, I have some additional remarks (listed below). They are very minor and mostly concern typos and grammar. Therefore, I'm not expecting a response at all and I am just listing them so the authors can make the appropriate corrections before sending the manuscript to the copy-editing team.

Good job!

**Minor comments**

General remark: there's a mix of nonlinear and non-linear

Abstract
L 20: **towards** as you're using British English

Introduction
L 56: References need to be in brackets
L 88: **(GIA)** not needed anymore since introduced earlier in introduction
L 96: same with **(GIA)**
L 99-100: it's four subsections now. Correct number and add sec 3.4

Method
L 116: has previously been applied
L 143: space missing between ° and **land**
L 156: not sure but space might be missing between **Fig.** and **1a**
L 181: as a consequence

Results
L 195-196: don't need the PI SMB value anymore here as it's been added to L194
L 201-202: This sentence is a bit as there as 2 references to the PI value
(**compared to the BG-PIcontrol run** and **above pre-industrial**)

L 217: 8 kyr or fewer

L 242: space missing between **Fig.** and **3**
L 243: 488±91 Gt/yr
L 243: twice as high as rather than two times higher than ?
L243-245: I think it would be better to separate into 2 sentences and remove the brackets. The sentence is already a bit long and there would be more emphasis on the fact that there is no « medium melt » anymore.
***We find that, when GIA is not accounted for, complete GrIS melt is achieved for an initial SMB forcing of 488±91 Gt/yr, which is twice as high as when GIA is included and corresponds to a global mean warming above pre-industrial of 1.6 K. The GrIS threshold behaviour is also exacerbated, with the final GrIS state switching directly from "low melt" to "complete melt" (Fig. 4).***

L 255-256: repeat of **in our simulations** makes it a bit more difficult to read. First one is not really necessary.
***As we do not account for ocean forcing at the marine-terminating outlet glacier, we triggered in our sensitivity simulations an increase in ice discharge …***

L 257: **(** instead of **,** before **see Subsection 2.3**
L 258: **,** after **(+3.2 K)**
L 269: Fig. 6 doesn't have the red box, only Fig. 7 has it. Better to add it to Fig. 6 than changing the reference though in this case
L284: remove **,**
L285: will or will not pass the threshold (remove extra **or not**)

Discussion
L 288: remove **we will**
L 288: **analyse** as you're using British English
L 300: **,** missing between **radiation** and **van Kampenhout**
L 304: remove **it** in and **it remains** (or needs to add something after analyse, i.e it remains complicated to analyse what?)
L 347: is also limited

L 349: has consistently been simulated
L 399: does not account